# Water Column Respiration in the Yakima River Basin is Explained by

# **Temperature, Nutrients and Suspended Solids.**

- Maggi M. Laan<sup>1+</sup>, Stephanie G. Fulton<sup>1+</sup>, Vanessa A. Garayburu-Caruso<sup>1\*</sup>, Morgan E. Barnes<sup>1</sup>, Mikayla A.
- Borton<sup>1,2</sup>, Xingyuan Chen<sup>1</sup>, Yuliya Farris<sup>1</sup>, Brieanne Forbes<sup>1</sup>, Amy E. Goldman<sup>1</sup>, Samantha Grieger<sup>3</sup>, Robert
- O. Hall, Jr.<sup>4</sup>, Matthew H. Kaufman<sup>1,5</sup>, Xinming Lin<sup>1</sup>, Erin L.M. Zionce<sup>1</sup>, Sophia A. McKever<sup>1</sup>, Allison Myers-
- Pigg<sup>3</sup>, Opal Otenburg<sup>3</sup>, Aaron C. Pelly<sup>1,6</sup>, Huiying Ren<sup>1</sup>, Lupita Renteria<sup>1</sup>, Timothy D. Scheibe<sup>1</sup>, Kyongho
- Son<sup>1</sup>, Jerry Tagestad<sup>1</sup>, Joshua M. Torgeson<sup>1</sup>, James C. Stegen<sup>1,6\*</sup>
- Pacific Northwest National Laboratory, Richland, Washington, USA
- <sup>2</sup>Colorado State University, Fort Collins, CO, USA
- <sup>3</sup>Pacific Northwest National Laboratory, Marine and Coastal Research Laboratory, Sequim, Washington, USA
- <sup>4</sup>Flathead Lake Biological Station, University of Montana, Polson, Montana, USA
- <sup>5</sup>Department of Earth, Environment, and Physics, Worcester State University, Worcester, Massachusetts, USA
  - <sup>6</sup>School of the Environment, Washington State University, Pullman, Washington, USA

13 14

- \*Correspondence to: Vanessa Garayburu-Caruso (vanessa.garayburu-caruso@pnnl.gov) and James C. Stegen
- (james.stegen@pnnl.gov)
- + The authors contributed equally

#### 18 Abstract

- Understanding aquatic ecosystem metabolism involves the study of two key processes: carbon fixation via primary production and
- organic C mineralization as total ecosystem respiration (ER $_{tot}$ ). In streams and rivers, ER $_{tot}$  includes respiration in the water column
- (ER<sub>wc</sub>) and in the sediments (ER<sub>sed</sub>). While literature surveys suggest that ER<sub>sed</sub> is often a dominant contributor to ER<sub>tot</sub>, recent
- studies indicate that the relative influence of sediment-associated processes versus water column processes can fluctuate along the
- river continuum. Still, a comprehensive understanding of the factors contributing to these shifts within basins and across stream
- orders is needed. Here we contribute to this need by measuring  $ER_{wc}$  and aqueous chemistry across 47 sites in the Yakima River
- $25 \qquad \text{basin, Washington, USA. We found that } ER_{wc} \text{ rates varied throughout the basin during baseflow conditions, ranging from } 0 \text{ to } -$
- $7.38 \text{ g O}_2 \text{ m}^{-3} \text{ d}^{-1}$ , and encompassed the entire range of  $ER_{wc}$  rates from previous work. Additionally, by comparing to  $ER_{tot}$
- estimates for rivers across the contiguous United States, we suggest that the contribution of ER<sub>wc</sub> rates to reach-scale ER<sub>tot</sub> rates
- across the Yakima River basin are likely highly variable, but we did not test this directly. We observed that ER<sub>wc</sub> is locally
- controlled by temperature, dissolved organic carbon, total dissolved nitrogen, and total suspended solids, which explained 40% of
- ER<sub>wc</sub> variability across the basin using Least Absolute Shrinkage and Selection Operator (LASSO) regression. Our findings
- highlight the potential relevance of water column processes in aquatic ecosystem metabolism across the entire stream network and
- that these influences are likely not predictable simply via position in the stream network. Our results are generally congruent with
- previous work in terms of locally-influential variables, suggesting that the observed variability and suite of associated
- environmental factors influencing ER<sub>wc</sub> are potentially transferable across basins.

## 1 Introduction

- Metabolism in streams and rivers includes both gross primary production (GPP) and ecosystem respiration (ER<sub>tot</sub>) as fundamental
- processes that shape energy dynamics and nutrient cycling in riverine systems (Bernhardt et al., 2018). GPP and ER<sub>tot</sub> impact

biogeochemical cycling through the fixation and subsequent breakdown and processing of carbon (C) in aquatic ecosystems (Allan et al., 2021; Genzoli & Hall, 2016; Hall, 2016; Hall & Hotchkiss, 2017; Reisinger et al., 2016). Riverine metabolism is modulated by various environmental features, including physical and biogeochemical factors. Physical parameters include discharge, flow regimes, flow extremes, light availability, and temperature (Bernhardt et al., 2022; Hensley et al., 2019; Jankowski & Schindler, 2019; Nakano et al., 2022). Biogeochemical influences include the availability, amount, and composition of C and other nutrients (Bertuzzo et al., 2022; Garayburu-Caruso et al., 2020b; Mulholland et al., 2008; Reisinger et al., 2021). Additionally, watershed characteristics such as stream size or drainage area, hydrologic connectivity, watershed geomorphology, and land use and land cover further affect these metabolic processes (Bernot et al., 2010; Demars, 2019; Finlay, 2011; Jankowski & Schindler, 2019).

Reach scale ecosystem metabolism encompasses biogeochemical processes that occur in both the water column and in benthic and hyporheic sediments (Hall & Hotchkiss, 2017). Historically, metabolism studies focused on headwater streams which are characterized by relatively large contact areas between surface water and the benthic sediments (Alexander et al., 2007; Battin et al., 2008; Findlay, 1995; Gomez-Velez et al., 2015; Mulholland et al., 2008; Peterson et al., 2001). Recent advances in computing power and the increased availability of high-resolution sensor data (e.g., dissolved oxygen, temperature, and river depth) have expanded the scope of metabolism studies beyond single small streams enabling researchers to investigate the relative contributions of ER<sub>sed</sub> and water column respiration (ER<sub>wc</sub>) to ER<sub>tot</sub> across diverse stream networks and orders. These efforts show that the proportion of ER<sub>tot</sub> derived from ER<sub>sed</sub> varies greatly across different sites, contributing from 3% to 96% of ER<sub>tot</sub> (Battin et al., 2003; Fuss & Smock, 1996; Gagne-Maynard et al., 2017; Jones Jr, 1995; Kaplan & Newbold, 2000; Naegeli & Uehlinger, 1997). This observed variability in the fraction of ER<sub>tot</sub> derived from ER<sub>sed</sub> indicates that ER<sub>wc</sub> may be important in certain places and times.

Water column processes, including nutrient cycling, occur at considerable rates and become increasingly important as rivers grow in size, marking a transition from benthic-dominated to water column-dominated processing (del Giorgio & Williams, 2005; Gardner & Doyle, 2018; Reisinger et al., 2015, 2016). Increases in downstream GPP (Finlay, 2011; Segatto et al., 2021), may influence ecosystem respiration, such that we would expect faster ER<sub>wc</sub> with greater GPP due to increases in C (Hall et al., 2016; Mejia et al., 2019). Additionally, greater N processing in the water column with increasing stream order (Wang et al., 2022), may suggest that water column biogeochemical processing increases along the stream network. Despite these trends, even as rivers increase in size, the relative contribution of ER<sub>wc</sub> to ER<sub>tot</sub> remains variable, likely in response to changing environmental conditions (Genzoli & Hall, 2016; Reisinger et al., 2021; Ward et al., 2018). This highlights a key knowledge gap that while the role of the water column in reach-scale processes such as GPP and ER<sub>tot</sub> likely fluctuates along the river network, this relationship remains poorly understood.

We contribute to addressing this knowledge gap by investigating the spatial variation of ER<sub>wc</sub> in the Yakima River basin, Washington, USA. The Yakima River basin is representative of the Columbia River basin, one of the largest river basins in the contiguous United States (CONUS), that spans the northwest region of CONUS. The Yakima River basin encompasses climatic regimes, biomes, physical settings, and land use conditions commonly found throughout the Columbia River basin and the western CONUS. Using the environmental diversity of the Yakima River basin, our goal was to generate knowledge of ER<sub>wc</sub> that could be transferable across the Columbia River basin and potentially beyond. We focus on ER<sub>wc</sub> during summer baseflow conditions and specifically 1) compare ER<sub>wc</sub> from the Yakima River basin to published ER<sub>wc</sub> and ER<sub>tot</sub> from other systems; 2) test the hypothesis that ER<sub>wc</sub> will be faster moving down the stream network; and 3) compare variables that explain variation in ER<sub>wc</sub> to those found

as explanatory in previous studies. To address these objectives, we estimated  $ER_{wc}$  and measured surface water chemistry at 47 sites across the Yakima River basin during the summer of 2021. Our estimates of  $ER_{wc}$  span all previously reported rates and while we did not observe faster  $ER_{wc}$  moving down the stream network, the most important explanatory variables did align with previous studies.

## 2 Methods

# 2.1 Methods Overview

Field sites in the Yakima River basin were selected to be representative of biophysical attributes of the larger Columbia River basin. For this, we grouped all catchments in the Columbia River basin into six classes sharing similar landscape characteristics using key biophysical attributes and selected sites in the Yakima River basin from each of the six classes. Final field locations spanned six Strahler stream orders and a wide range of land cover types and physical settings. We used dark bottle incubations and collected surface water chemistry samples to study the spatial variability of ER<sub>wc</sub> at a basin scale with respect to environmental conditions during summer baseflow conditions in 2021. We also compared ER<sub>wc</sub> observed in the Yakima River basin against literature ER<sub>wc</sub> and ER<sub>tot</sub> values to understand how the Yakima River basin relates to streams and rivers across the world. We used Least Absolute Shrinkage and Selection Operator (LASSO) regression to evaluate the relationship between ER<sub>wc</sub> and drainage area, stream temperature, surface water chemistry, and organic matter putative biochemical transformations as a proxy for the diversity of reactions occurring in upstream reaches to determine the primary factors influencing ER<sub>wc</sub> throughout the Yakima River basin. All analyses were performed using R Statistical Software (v4.2.0). All data generated from the sampling study, including data not evaluated in this manuscript, are publicly available.

#### 2.2 Watershed characterization and site selection

- The Yakima River basin is the fifth-largest basin in the Columbia River basin and is located entirely within the state of Washington,
- USA. The basin is roughly 16,000 km<sup>2</sup> and spans forested mountainous regions in the west to arid valleys and plains in the east.
- The basin has a diversity of land covers and land uses dominated by shrubland, forest, and agriculture. Annual precipitation ranges
- from up to 350 cm in the west to 25 cm in the east (Vano et al., 2010).

To enable further testing of the transferability of study results to catchments throughout the Columbia River basin, we strategically selected sampling sites in the Yakima River basin based on their biophysical (e.g. hydrology, topography, vegetation type) characteristics. This was done by first grouping all National Hydrography Dataset Plus Version 2.1 (NHDPlusV2.1) catchments (McKay et al., 2012) in the Columbia River basin (n = 181,531) into six classes sharing similar landscape characteristics using cluster analysis. To capture the variability in biophysical settings found across the Columbia River basin, we selected 16 key attributes as input variables to the cluster analysis, including climate, vegetation structure and function, topography, and wildfire potential (Table S1). We then selected multiple sites within each of the six Columbia River basin classes. Existing, readily available geospatial data came from multiple sources including NASA Moderate Resolution Imaging Spectroradiometer (eMODIS) Remote Sensing Phenological (RSP) data (U. S. Geological Survey, 2019); NASA MODIS land cover type (Friedl & Sulla-Menashe, 2019); NASA MODIS normalized difference vegetation index (NDVI), fraction of photosynthetically active radiation (FPAR, %), and leaf area index (LAI,  $m^2 m^{-2}$ ) (Myneni et al., 2015); NASA MODIS total evapotranspiration (ET, kg H<sub>2</sub>O  $m^{-2}$  d<sup>-1</sup>) (Running et al., 2017); NASA MODIS terrestrial net primary productivity (NPP, kg C  $m^{-2}$  y<sup>-1</sup>) and terrestrial net ecosystem productivity data (NEP, kg C  $m^{-2}$  y<sup>-1</sup>) (Running & Zhao, 2019); PRISM precipitation data (PRISM Climate Group, Oregon State University,

2023); NHDPlusV2.1 stream length and catchment boundaries (McKay et al., 2012); USGS National Elevation Dataset (NED) 1/3 Arc-Second Digital Elevation Model topography data (U.S Geological Survey, 2023); USFS Wildfire Hazard Potential (WHP) data (Dillon, 2018); and Landscape Fire and Resource Management Planning Tools (LANDFIRE) existing vegetation percent cover (%) and height (m) data (Dillon & Gilbertson-Day, 2020).

118119120

We used a k-means clustering algorithm using the kmeans function within the 'stats' package in base R to group NHDPlusV2.1 catchments with similar properties using the normalized, statistical moments (minimum, maximum, mean, and standard deviation (SD)) for 70 geospatial variables within each NHDPlusV2.1 catchment (Table S1) as input. To calculate statistical moments for each variable, we summarized geospatial data types at the NHDPlusV2.1 catchment level using two different methods: zonal statistics for continuous raster data and tabulation for vector data. Zonal statistics calculate statistical moments by individual catchment polygon. Tabulation calculates total length or area of a particular vector feature within each individual catchment polygons. We evaluated 13 different sets of variable-statistical moment combinations for use in the cluster analysis and selected variable set 8, which included the zonal mean and zonal standard deviation for 70 variables (n = 140) (Table S2). Once the data for variable set 8 were summarized at the NHDPlusV2.1 catchment level, we calculated z-scores (z) for each geospatial variable. Resultant z-scores for variable set 8 were fed into the k-means classifier, which iteratively adds each catchment to one of n clusters, with n being set by the user (n = 15, this study), using Euclidean distance to minimize within-cluster distance and maximize between-cluster distance. We ran multiple iterations of the cluster analysis using 2-15 clusters using the mean and standard deviation of all variables. To visualize the reduction in within-cluster variation between iterations 1–15, we generated elbow plots by plotting the Within Cluster Sum of Squares (WCSS) value against the total number of catchments in a cluster and selected six clusters as the suitable number of clusters that minimized map visual complexity enough to guide manual site selection while maintaining a level of variation in key biophysical characteristics representative of the Columbia River basin. Clusters 1 and 3–6 were categorized according to tree height, precipitation, and elevation (Table 1 and Table S3). Cluster 2 was categorized as "Water dominated" and was not used for selecting sites. Cluster analysis results were then used to guide the selection of 47 field sites distributed across Strahler stream orders 2-7 (the highest order stream in the Yakima River basin) that spanned the basin and captured the variation in biophysical characteristics represented by clusters 1 and 3–6 (Fig. S1). First order and other non-perennial streams were not sampled due to the lack of flow during summer baseflow or baseflows were too low to support sampling. We attempted to include logistical considerations in model-based site selection, but this task proved impractical and field-scouting trips were needed to refine site selections. Day-of-sampling changes to the sampling plan were made on-the-fly when the Schneider Springs Fire started on the Okanogan-Wenatchee National Forest. Fire activity and road closures restricted access to a large portion of the Yakima River basin, primarily in the Tieton River and American River watersheds located in the midwestern portion of the basin.

Table 1. Cluster analysis results characterizing NHDPlusV2.1 catchments across the Columbia River basin and Yakima River basin with similar biophysical and hydrologic characteristics and the number and percentage of sites in each basin.

| Cluster | Name                            | CRB      | YRB      | YRB Sites | Percent     |
|---------|---------------------------------|----------|----------|-----------|-------------|
|         |                                 | Drainage | Drainage | Per       | YRB Sites   |
|         |                                 | Area     | Area     | Cluster   | Per Cluster |
| 1       | Tree dominated high elevation   | 23%      | 27%      | 9         | 19%         |
|         | mesic                           |          |          |           |             |
| 2       | Water dominated                 | 3%       | 2%       | 0         | 0%          |
| 3       | Tree dominated high elevation   | 7%       | 2%       | 2         | 4%          |
|         | hydric                          |          |          |           |             |
| 4       | Shrub-steppe middle elevation   | 25%      | 28%      | 10        | 21%         |
|         | xeric                           |          |          |           |             |
| 5       | Tree dominated middle elevation | 17%      | 17%      | 13        | 28%         |
|         | mesic                           |          |          |           |             |
| 6       | Tree dominated middle elevation | 24%      | 23%      | 13        | 28%         |
|         | xeric                           |          |          |           |             |

"CRB Drainage Area" is the percentage of the total drainage area of the Columbia River basin that was classified in each cluster. "YRB Drainage Area" is the percentage of the total drainage area of the Yakima River basin that was classified in each cluster. "YRB Sites Per Cluster" is the total number of field sites in the Yakima River basin (n = 47) located in each cluster. "Percent YRB Sites Per Cluster" is the percentage of the total number of sampling sites in the Yakima River basin located in each cluster.

#### 2.3 Water column respiration data collection

We measured  $ER_{wc}$  (g  $O_2$  m<sup>-3</sup> d<sup>-1</sup>) in triplicate for 2 h at each site between 30 August and 15 September 2021 using a modified "semi-*in situ*" dark bottle incubation (Genzoli & Hall, 2016) (Fig. 1a). Calibrated dissolved oxygen (DO) sensors (miniDOT Logger; Precision Measurement Engineering, Inc.; Vista, CA, USA) recorded DO concentration (mg L<sup>-1</sup>) and temperature (°C) at 1 min intervals in 2-L dark bottles (Nalgene<sup>TM</sup> Rectangular Amber HDPE bottles; ThermoFisher Scientific; Waltham, Massachusetts, USA) (Fulton et al., 2022). Bottle necks were slightly widened (1 to 2 mm) to accommodate the diameter of the DO sensor.

At the start of each sampling day, DO sensors and all sampling equipment were placed in a cooler with blue ice packs to keep them cool and minimize the time needed at each site for the sensors to equilibrate with the similarly cool river water temperatures. Upon arrival at each site, bottles were rinsed three times with river water and then filled by wading as close to the thalweg as possible, submerging the bottles below the river surface, and rolling them 360 degrees while held upright underwater to ensure no air bubbles were present in the bottles (Fig. 1a). Bottles were secured upright in a cooler filled with river water, placed in the shade on the streambank, and allowed to equilibrate for 20 min. Following the 20 min equilibration period, the bottles were emptied and refilled with fresh river water and a small, battery-powered mixing device (Underwater Motor, Item Number 7350; Playmobil; Shanghai, China; rechargeable AA NiMH battery; Amazon; Seattle, Washington, USA) and the DO sensor was gently inserted (sensor face-up) in the bottles to minimize trapping air bubbles in the bottles. The bottles were capped underwater and returned to the water-filled cooler. The bottles were incubated for 2 h, and river water surrounding the bottles in the cooler was replenished every 20 min to maintain *in situ* temperature.

173174

192193

**Figure 1. Modified semi-***in situ* dark bottle incubation method and example study sites. (a) Underwater photograph of DO sensor being inserted into an incubation bottle filled with river water and mixing device. Right panels emphasize the diversity of environmental settings covered in this study. (b) North Fork Teanaway River (site S19E), Kittitas County, Washington, September 2021. Site S19E is classified as a mesic, high elevation site dominated by tree canopy (Cluster 1; see Table 1, Table S3, Fig. S1). (c) Yakima River at Mabton (site T02), Yakima County, Washington, September 2021. Site T02 is classified as a mesic, middle elevation site dominated by tree canopy (Cluster 5; see Table 1, Table S3, Fig. S1).

# 2.4 Surface water chemistry sample collection and analysis

Filtered surface water samples were collected at each site for dissolved inorganic C (DIC, mg  $L^{-1}$ ); dissolved organic C (DOC, mg L<sup>-1</sup>); total dissolved N (TDN, mg L<sup>-1</sup>); anions, including nitrate (NO<sub>3</sub><sup>-</sup>, mg L<sup>-1</sup>), chloride (Cl<sup>-</sup>, mg L<sup>-1</sup>), and sulfate (SO<sub>4</sub><sup>2</sup><sup>-</sup>, mg L<sup>-1</sup>); and DOM chemistry using a 50-mL syringe and 0.22 µm sterivex filter (MilliporeSigma<sup>TM</sup> Sterivex<sup>TM</sup> Sterile Pressure-Driven Devices; MilliporeSigma<sup>TM</sup>; Burlington, Massachusetts, USA) (Grieger et al., 2022). Samples were collected in triplicate from 50% of the water column depth. Prior to sample collection, filter assemblies were rinsed once by pushing 5 mL of river water through the filter. DIC, DOC and TDN samples were filtered into 40 mL amber glass vials (Amber Clean Snap Vials; Thermo Fisher Scientific; Waltham, Massachusetts, USA). DIC samples were collected by attaching a sterile 18 g needle (BD General Use and PrecisionGlide Hypodermic Needles; Becton, Dickinson and Company; Franklin Lakes, NJ, USA) to the sterivex filter and pushing three vial-volumes of river water (~150 mL) slowly through the syringe to prevent the introduction of air bubbles to the sample, allowing the vials to overflow continuously. When the final 50 mL of river water was pushed through the syringe, the vials were capped with a surface tension dome of water to ensure no headspace. Samples collected for ion analysis were filtered into a 15 mL conical tube (Olympus<sup>TM</sup> Plastics; Genesee Scientific; Morrisville, NC, USA). Samples collected for DOM chemistry were filtered into pre-acidified (85 % phosphoric acid, H<sub>3</sub>PO<sub>4</sub>) 40 mL amber vials (Amber Clean Snap Vials; Thermo Fisher Scientific; Waltham, Massachusetts, USA) (Grieger et al., 2022). One unfiltered grab sample for total suspended solids (TSS, mg L<sup>-1</sup>) was collected using a pre-washed 2-L amber bottle (Nalgene<sup>TM</sup> Rectangular Amber HDPE Bottles; ThermoFisher Scientific; Waltham, Massachusetts, USA). TSS bottles were rinsed three times with river water prior to sample collection. All samples were stored on ice in the field and then refrigerated at 4° C before shipping for analysis to the Pacific Northwest National Laboratory (PNNL) Marine and Coastal Research Laboratory in Sequim, Washington (DOC and DIC) and PNNL Biological Sciences Facility

Laboratory in Richland, Washington (TSS, ions, and DOM). TSS samples were analyzed within one week of collection, DOC and TDN were measured within two weeks of collection, DIC was measured within one month of collection, and ion and DOM samples were frozen (-20 °C) upon receiving until analysis.

DOC, TDN, and DIC were measured on a Shimadzu TOC-L Total Organic Carbon Analyzer. DOC was measured as non-purgeable organic C (NPOC). Anion concentrations were determined quantitatively on a Dionex ICS-2000 anion chromatograph with AS40 autosampler using one replicate. An isocratic method was used with 23 mM KOH eluent at 1 mL/minute at 30°C. The analytical column was an IonPac AS18 (4 x 250 mm, Dionex catalog # 060549). The suppressor was a ADRS 600 set at 57 mA (4 mm, self regenerating, Dionex catalog # 088666). Concentrations below the limit of detection (LOD) of the instrument, or below the standard curve, were flagged (Grieger et al., 2022). For other samples below the lowest standard value (TDN: 0.1 mg L<sup>-1</sup>, NO<sub>3</sub>: 0.07 mg L<sup>-1</sup>), one half of the lowest standard value was used (TDN: 0.05 mg L<sup>-1</sup>, NO<sub>3</sub>: 0.035 mg L<sup>-1</sup>) for statistical analysis. For samples below the limit of detection (TDN LOD: 0.07 mg L<sup>-1</sup>; NO<sub>3</sub>: LOD: 0.07 mg L<sup>-1</sup>), but above the lowest standard, one half of the LOD value (TDN: 0.035 mg L<sup>-1</sup>; NO<sub>3</sub>: 0.035 mg L<sup>-1</sup>) was used for analysis. Phosphate (PO<sub>4</sub><sup>3-</sup>) was measured, however, over two thirds of samples showed values below detection, and thus the analyte was not used in subsequent analyses. Pairwise differences between NPOC, TDN, and DIC measurements from all replicates were calculated. The sample that had the largest difference from the other samples was removed if the coefficient of variation was greater than 30%. This coefficient of variation threshold for sample removal is based on inspecting histograms of these data types, and determining the point at which sites likely contain anomalous outlier values. Parameter mean values for each site were then calculated from the remaining replicates.

TSS samples were filtered in the laboratory through a pre-weighed and pre-combusted 4.7 cm, 0.7 µm GF/F glass microfiber filter (Whatman<sup>TM</sup> glass microfiber filters, Grade 934-AH®; MilliporeSigma; Burlington, Massachusetts, USA). After water filtration, the filter and filtration apparatus were rinsed with 30 mL of ultrapure Milli-Q water (Milli-Q® IQ Water Purification System; MilliporeSigma; Burlington, Massachusetts, USA) to ensure that all residue was captured by the filter. The filter was placed in foil and oven dried overnight at 45° C. TSS (mg L<sup>-1</sup>) was calculated as the difference between the weight (mg) of the filter before and after filtration of the water sample divided by the volume of water filtered (L). For samples below the LOD, one half of the LOD value (LOD: 0.24 mg L<sup>-1</sup>) was used for analysis.

## 2.5 DOM chemistry via ultra-high resolution mass spectrometry and biochemical transformations

Organic matter chemistry was characterized via ultra-high resolution mass spectrometry using a 12 Tesla (12T) Bruker SolariX Fourier transform ion cyclotron resonance mass spectrometer (FTICR-MS) at the PNNL Environmental Molecular Sciences Laboratory in Richland, Washington, following methods described in Garayburu-Caruso et al. (2020a). Measured DOC concentrations were used to normalize the DOC concentration of the sample to 1.5 mg C L<sup>-1</sup> prior to further processing. Samples were thawed in the dark at 4°C overnight before acidifying to pH 2 using 85 % H<sub>3</sub>PO<sub>4</sub>. Samples were then subjected to solid phase extraction (SPE) using Bond Elut PPL cartridges (Agilent; Santa Clara, CA, USA) following protocols employed by Dittmar et al. (2008). Extracted samples were run in the FTICR-MS with a standard electrospray ionization source in negative mode. Data were collected with an ion accumulation time of 0.08 seconds. BrukerDaltonik Data Analysis version 4.2 was used to convert raw spectra to a list of molecular compound mass-to-charge ratios (m/z) with a signal-to-noise ratio (S/N) threshold set to 7 and absolute intensity threshold to the default value of 100. Peaks were aligned (0.5 ppm threshold) and molecular formula were assigned using the Formularity software with S/N > 7 and mass measurement error < 0.5 ppm (Tolić et al., 2017). The Compound Identification algorithm takes into consideration the presence of C, H, O, N, S, and P and excludes other elements. Aligned and calibrated data

was further processed using ftmsRanalysis (Bramer et al., 2020). Replicate samples were merged into one site where peaks in a sample were retained if they were present in at least one of the replicates. DOM biochemical transformations were inferred following methods previously employed by Ryan et al., (2024); Danczak et al., (2023); Fudyma et al., (2021); Garayburu-Caruso et al., (2020); Stegen et al., (2018). In summary, we calculated pairwise mass differences between every peak in a sample regardless of molecular formula assigned and compared that mass difference to a list of 1,255 molecular masses associated with commonly observed biochemical transformations (Table S4). Biochemical transformations allow you to infer the number of times the mass that corresponds to a specific molecule is gained or lost. For example, if a mass difference between two peaks corresponded to 128.095, that would correlate to the loss or gain of the amino acid lysine (see Table S4). We further calculated the total number of DOM transformations per site and the total number of DOM transformations normalized by the number of peaks present in the site (i.e., "normalized DOM transformations").

## 2.6 DO sensor data cleaning, processing, and analysis

We extracted the raw DO concentration (mg  $O_2$  L<sup>-1</sup>) and temperature (°C) sensor data for each site and plotted DO and temperature against incubation time for each set of triplicate incubations (n = 141). The plots were visually inspected to a) confirm that temperature sensors were at equilibrium with the river temperature when the 2 h incubation test period began and b) identify data gaps, outliers, and other data anomalies. Following the visual inspection of plots, the first 5 min of the time series was removed, then the data was trimmed to 90 min to account for anomalies due to emptying and refreshing river water in the bottles, and to ensure all sites had the same incubation time. Sensor data distributions were also evaluated using violin plots for each site.

 $ER_{wc}$  rates for individual triplicate incubation samples were calculated as the slope of the linear regression between the DO sensor data and the incubation time, which was converted to units of g  $O_2$  m<sup>-3</sup> d<sup>-1</sup>. All samples met the normalized root mean square error (NRMSE) criteria of  $\leq 0.01$  (Shcherbakov et al., 2013). Mean  $ER_{wc}$  for each site and the global mean and variance were then calculated from the samples (n = 141). Nearly one-fifth of  $ER_{wc}$  values were slightly positive. Positive respiration rates are biologically unrealistic, however positive values less than 0.5 g  $O_2$  m<sup>-3</sup> d<sup>-1</sup> are difficult to distinguish from zero (Appling et al., 2018b). Thus, we changed positive  $ER_{wc}$  values less than 0.5 g  $O_2$  m<sup>-3</sup> d<sup>-1</sup> to 0 for analysis and removed values greater than 0.5 g  $O_2$  m<sup>-3</sup> d<sup>-1</sup> (n = 2).  $ER_{wc}$  values greater than 0.5 g  $O_2$  m<sup>-3</sup> d<sup>-1</sup> were observed when the DO concentration in the bottle started near 5 mg  $O_2$  L<sup>-1</sup> and increased over the 2-hour incubation period. The increase in concentration and the high, positive respiration rate is likely due to diffusion of DO through the bottle walls.  $ER_{wc}$  values are reported in volumetric units (g m<sup>-3</sup> d<sup>-1</sup>) as opposed to areal units (g m<sup>-2</sup> d<sup>-1</sup>) due to difficulties in obtaining high quality depth data across all field sites, spanning small headwater streams to large mainstem rivers.

## 2.7 Relationship of water column respiration rates to watershed characteristics and surface water chemistry

We evaluated the relationship between  $ER_{wc}$ , watershed characteristics, physical parameters, and surface water chemistry using LASSO regression models, which perform variable selection and model regularization, to establish the suite of explanatory variables that most influence variation in  $ER_{wc}$  across the Yakima River basin. We observed that several model input variables had skewed distributions, thus a cube root transformation was applied to all variables to reduce the impact of high leverage points in the regression analysis. Further, all data was standardized as z-scores before analysis to ensure all data was in the same quantitative range.  $\beta$  coefficients reported for each variable were calculated by performing LASSO regression using the *glmnet* function in R (Friedman et al., 2010) over 100 random seeds, normalizing to the maximum  $\beta$  coefficient in each regression, and averaging the normalized  $\beta$  coefficients across the 100 iterations. The minimum penalty parameter ( $\lambda$ ) determined by cross validation was used

in each regression. Because LASSO regression was used for exploratory analysis, not prediction, data was not split into training and testing sets. LASSO does not inherently estimate  $R^2$ , so we calculated it using the total sum of squares and residual sum of squares for each fitted model, as traditionally done with standard multiple regression. The estimation of residual sum of squares used predicted values of  $ER_{wc}$  based on the explanatory variables used in the model. The  $R^2$  estimates were used to estimate how much variation in  $ER_{wc}$  was explained by each of the LASSO models. Standard deviation of  $\beta$  coefficients were compared to mean values of  $\beta$  coefficients to confirm that the most important variables were relatively consistent across seeds. Total drainage area  $ER_{wc}$  was defined as the total upstream drainage area from each site and was extracted for each site from the NHDPlusV2.1 stream database using site latitude and longitude. Stream order for each site was extracted as the reach attribute "StreamOrde" from the NHDPlusV2.1 stream database, which is a modified version of Strahler stream order (Blodgett & Johnson, 2022; McKay et al., 2012; Willi & Ross, 2023). To evaluate whether the directionality of relationships observed in the LASSO regression were consistent with univariate relationships, we used Pearson correlations between  $ER_{wc}$ , drainage area, water chemistry, and environmental factors; these correlations were calculated using the *cor* function in R.

## 2.8 Comparison to published water column respiration rates

To contextualize the magnitude of observed  $ER_{wc}$  rates in the Yakima River basin, we compared our results to published literature values of  $ER_{wc}$  (n = 118) (Table S5) and  $ER_{tot}$  (n = 208). Published  $ER_{wc}$  values were converted to volumetric units (g  $O_2$  m<sup>-3</sup> d<sup>-1</sup>) using standard unit conversions. For example, molar values ( $\mu$ mol  $O_2$  L<sup>-1</sup> H<sup>-1</sup>) as in Devol et al. (1995) and Quay et al. (1995) were corrected using the molar mass of oxygen and standard time conversions. When  $ER_{wc}$  was reported with respect to C or carbon dioxide (CO<sub>2</sub>), as in Ellis et al. (2012) and Ward et al. (2018), conversions provided in the text were used to convert to an  $O_2$  basis. Areal estimates of  $ER_{wc}$  (g  $O_2$  m<sup>-2</sup> d<sup>-1</sup>), as in Genzoli and Hall (2016) and Reisinger et al. (2021), were converted to volumetric units by multiplying by 1/depth (m<sup>-1</sup>) using same-day depth data for each reach studied. We also compared our  $ER_{wc}$  values to daily reach-averaged estimates of  $ER_{tot}$  (n = 490,907) for 356 rivers and streams across the CONUS by using the datasets published in Appling et al., (2018b) and Bernhardt et al., (2022) where  $ER_{tot}$  was estimated by a single-station, open channel approach using the streamMetabolizer package in R (Appling et al., 2018b, 2018a). For our comparative analysis, we used the cleaned, gap-filled data from Bernhardt et al. (2022) (n = 208). The Bernhardt et al. (2022) sites are a subset from the Appling et al. (2018a, 2018b) dataset generated through a robust data quality analysis to remove sites potentially affected by process or observation error. For comparison with our  $ER_{wc}$  values, we first averaged Bernhardt et al. (2022)  $ER_{tot}$  areal units (g  $O_2$  m<sup>-2</sup> d<sup>-1</sup>) at each site. Then, average  $ER_{tot}$  values were converted to volumetric units by calculating average river depth per site from the Appling et al. (2018a, 2018b) dataset and multiplying average  $ER_{tot}$  by 1/depth.

## 3 Results and discussion

## 3.1 Yakima River basin ER<sub>wc</sub> rates spanned literature values

At baseflow conditions,  $ER_{wc}$  varied widely across the Yakima River basin. The linear regression models for each triplicate set of DO sensor measurements were well-fit to the data and all sites met the criteria for NRMSE  $\leq$  0.01 (Fig. S2; Fig. S3). We observed consistency across triplicate measurements, illustrating that the method was effective in providing repeatable estimates of  $ER_{wc}$  rates throughout the Yakima River basin (Fig. S2; Fig. S3). After removing positive respiration rates > 0.5 g  $O_2$  m<sup>-3</sup> d<sup>-1</sup>, which were associated with diffusion effects on DO, and turning small positive rates to zero,  $ER_{wc}$  rates ranged from -7.38 to 0 g  $O_2$  m<sup>-3</sup> d<sup>-1</sup>, with a median value of -

 $0.58 \text{ g O}_2 \text{ m}^{-3} \text{ d}^{-1} \text{ (mean: -0.84 g O}_2 \text{ m}^{-3} \text{ d}^{-1}, \text{ standard deviation} = 1.23 \text{ g O}_2 \text{ m}^{-3} \text{ d}^{-1} \text{) (Fig. 2a)}.$ 

315316

319320

The values of ER<sub>wc</sub> observed in our study spanned the range of published literature values (Fig. 2; Table S5). From 118 published measurements of ER<sub>wc</sub> across the CONUS and the Amazon River basin, ER<sub>wc</sub> ranged from -4.63 g O<sub>2</sub> m<sup>-3</sup> d<sup>-1</sup> to -0.07 g O<sub>2</sub> m<sup>-3</sup> d-1. We compared median values, rather than means, across studies as medians are more appropriate for skewed distributions and are less sensitive to outliers in the data. The median ER<sub>wc</sub> from this study (-0.58 g  $O_2$  m<sup>-3</sup> d<sup>-1</sup>) is slower than the median of literaturereported ER<sub>wc</sub> values (-0.96 g O<sub>2</sub> m<sup>-3</sup> d<sup>-1</sup>). However, the fastest ER<sub>wc</sub> rate in the Yakima River basin (-7.38 g O<sub>2</sub> m<sup>-3</sup> d<sup>-1</sup>), exceeded the fastest reported literature value (-4.63 g O<sub>2</sub> m<sup>-3</sup> d<sup>-1</sup>) (Reisinger et al., 2021). Reisinger et al. (2021) measured ER<sub>wc</sub> in 15 midsized rivers across basins with differing turbidity levels and nutrient concentrations, finding a similar median ER<sub>wc</sub> (-0.60 g O<sub>2</sub> m<sup>-3</sup> d<sup>-1</sup>) to this study. In the Klamath River, median ER<sub>wc</sub> (-0.51 g O<sub>2</sub> m<sup>-3</sup> d<sup>-1</sup>) was also similar to the Yakima River basin. However, ER<sub>wc</sub> doubled following summer cyanobacteria blooms, emphasizing the temporal variability in water column processes with changing environmental conditions (Genzoli & Hall, 2016). In the Amazon basin, literature comparisons varied, with median ER<sub>wc</sub> measurements similar to those found in the Yakima River basin in some studies (Devol et al., 1995; Ellis et al., 2012; Quay et al., 1995) and faster than the Yakima River basin in others (Ward et al., 2018). Ward et al. (2018) highlighted the importance of mixing in large rivers, noting that previous measurements of aquatic respiration in large tropical rivers, such as those measured in Quay et al. (1995) and Devol et al. (1995), may underestimate microbial respiration contribution due to lack of mixing during rate measurements. While comparisons across study medians are variable, the observation that ER<sub>wc</sub> in the Yakima River basin spans — and exceeds — reported literature values highlights the potential for using it as a test basin for understanding and uncovering transferable principles linked to stream metabolism.

330331332

346347

While ERtot estimates are not available across the Yakima River basin at the time of ERwc estimation for this manuscript, measured ER<sub>wc</sub> rates spanned a large fraction of CONUS-scale ER<sub>tot</sub> rates estimated by Appling et al., (2018a, 2018b) and Bernhardt et al. (2022). ERtot rates are reach-scale estimates of stream metabolism derived from time series measurements of DO. This method assumes well-mixed conditions such that sensor measurements represent homogenous reach observations. Under well-mixed conditions, ER<sub>wc</sub> measurements from dark bottle incubations are also representative of reach-scale processes (Genzoli & Hall, 2016). The median ER<sub>tot</sub> for 208 CONUS measurements was -5.25 g  $O_2$  m<sup>-3</sup> d<sup>-1</sup> with a range from -36.55 to -3.73 g  $O_2$  m<sup>-3</sup> d<sup>-1</sup>. The median ER<sub>wc</sub> rate (-0.58 g O<sub>2</sub> m<sup>-3</sup> d<sup>-1</sup>) observed in the Yakima River basin was 11% of median ER<sub>tot</sub> (Fig. 2). The fastest ER<sub>wc</sub> rate in the Yakima River basin (-7.38 g O<sub>2</sub> m<sup>-3</sup> d<sup>-1</sup>), was faster than the median ER<sub>tot</sub> (Fig. 2). While both ER<sub>tot</sub> and ER<sub>wc</sub> measurements span a range of stream conditions, we acknowledge that we did not compare these rates directly at the same places and times. However, given the overlap of ER<sub>wc</sub> from the Yakima River basin with CONUS-scale ER<sub>tot</sub>, we suggest that ER<sub>wc</sub> could typically represent a small fraction of ERtot but may occasionally have larger contributions across the Yakima River basin. If we had observed consistently very slow ER<sub>wc</sub> across the Yakima River basin, there would be little overlap with literature ER<sub>tot</sub> values, and we would have inferred consistently small contributions of ER<sub>wc</sub> to ER<sub>tot</sub>. In comparison, Genzoli and Hall (2016) observed that before summer cyanobacteria blooms, ER<sub>wc</sub> contributed around 10% of ER<sub>tot</sub> in sites along the Klamath River, with the contribution of ER<sub>wc</sub> to ER<sub>tot</sub> increasing following cyanobacteria blooms. Additionally, Reisinger et al. (2021) found that ER<sub>wc</sub> was not the dominant contributor to ERtot in mid-sized rivers, except at sites with low ERtot (mean ERwc contributions to ERtot: 35%, range 2 -81%). While these studies have shown spatiotemporal variability of the contributions of ER<sub>wc</sub> to ER<sub>tot</sub>, exploring these relationships in the Yakima River basin requires further research where ERtot is measured in conjunction with ERwc.

Figure 2. Water column respiration data from the Yakima River basin ( $ER_{wc}$  (this study); n=45), published water column respiration rates ( $ER_{wc}$  (Lit); n=118), and reach-scale estimates of ecosystem respiration by Appling et al., (2018a, 2018b) and Bernhardt et al. (2022) ( $ER_{tot}$ ; n=208). (a) Kernel density plots of  $ER_{wc}$  from the Yakima River basin (this study), published  $ER_{wc}$  rates (Lit) that have been converted to the same units as this study ( $g O_2 m^3 d^{-1}$ ), and published reach-scale  $ER_{tot}$  (Lit) from Bernhardt et al. (2022) that have been converted to volumetric units using depth data from Appling et al. (2018a). The left y-axis is for  $ER_{wc}$  values. The right y-axis is for  $ER_{wc}$  observed in the Yakima River basin (-0.58 g  $O_2 m^3 d^{-1}$ ). The vertical red line is the median  $ER_{wc}$  values from studies in rivers across the CONUS and the Amazon River basin (-0.96 g  $O_2 m^3 d^{-1}$ ). The vertical black line is the median  $ER_{wc}$  value (-5.25 g  $O_2 m^3 d^{-1}$ ). (b) Boxplots of published  $ER_{wc}$  and  $ER_{wc}$  from the Yakima River basin. The blue horizontal dashed line represents median  $ER_{wc}$  in the Yakima River basin. The red horizontal dashed line represents median  $ER_{wc}$  from literature values.

## 3.2 Water column respiration rates varied weakly with drainage area and stream order

 We observed a correlation between ER<sub>wc</sub> and drainage area across the Yakima River basin that was weak enough that we consider it inconsistent with our hypothesis that ER<sub>wc</sub> is faster moving down the stream network (Fig. 3). In latter sections, we use multivariate analysis for further evaluation of the relationships between ER<sub>wc</sub> and explanatory variables. The lack of a strong connection between ER<sub>wc</sub> and drainage area is somewhat surprising as such a relationship could emerge from downstream C transport as well as increasing autochthonous C inputs due to increasing temperature and light availability, providing additional substrate for microbial respiration (Finlay, 2011; Webster, 2007). The fastest observed ER<sub>wc</sub> rate in the Yakima River basin occurred in an agriculturally influenced, low gradient, 5th order stream, as opposed to our hypothesis of ER<sub>wc</sub> being fastest in the highest stream orders (Fig. 3). The conditions at this sampling location were not representative of the whole drainage area, as areas upstream of this site are mountainous with little human influence. This finding suggests that localized factors, not upstream conditions or drainage area, provide primary controls over ER<sub>wc</sub>. Anthropogenic impacts, such as from agriculture and urbanization, can alter nutrient dynamics and flow regimes in these areas, influencing biogeochemical processes such as ER<sub>wc</sub> (Bernot et al., 2010). Additionally, while we report ER<sub>wc</sub> on a volumetric basis, we acknowledge that this approach does not account for variation in water column depth along the river continuum. As river depth increases downstream, we expect the areal contribution of water column processes will also increase because areal contributions integrate across the whole water column (Wang et al., 2022). The weak correlation between volumetric-based ER<sub>wc</sub> and drainage area in the Yakima River basin likely reflects the interplay of multiple factors, including spatially variable local conditions, underscoring the complex controls on ecosystem processes in this region.

**Figure 3.** ER<sub>wc</sub> across the Yakima River basin and its relationship with total drainage area. (a) Map of land use/land cover classes in the Yakima River basin with ER<sub>wc</sub> values (g O<sub>2</sub> m<sup>-3</sup> d<sup>-1</sup>) overlaid. Faster rates are indicated by larger circle diameters. The fastest rate is indicated by the yellow circle. The map was generated using the Free and Open Source QGIS (v. 3.16.1 and v. 3.26.0). Map data include catchment boundaries and hydrography from the National Hydrography Dataset Plus (NHDPlusV2.1) (McKay et al., 2012) and 2016 land use/land cover data from the National Land Cover Dataset (Brown, 2024). (b) Scatter plot of cube root transformed ER<sub>wc</sub> related to cube root transformed total drainage area with points colored by stream order. The Pearson correlation coefficient (r) is provided on the panel.

## 3.3 Higher temperatures and nutrient concentrations are associated with faster ERwc.

Regression analyses showed that  $ER_{wc}$  in the Yakima River basin varied with chemical and physical water quality parameters. TDN, temperature, DOC, and TSS emerged as key variables in the LASSO regression, whereby  $ER_{wc}$  was faster with higher values of all these variables (Table 2). The LASSO regression explained 40% of the variation in  $ER_{wc}$  (Table 2). LASSO results are similar to univariate relationships, whereby DOC, TDN, temperature, and TSS had the strongest correlations with  $ER_{wc}$  (r = -0.46 to -0.63) (Fig. 4, Fig. S4) and all correlations were qualitatively in the same direction as indicated by the LASSO  $\beta$  coefficients. Changing positive  $ER_{wc}$  values less than 0.5 g  $O_2$  m<sup>-3</sup> d<sup>-1</sup> to 0 did not change the overall interpretation of univariate or multivariate relationships (Fig. S4, Fig. S5, Table S4). Collectively, the relative importance of these variables suggests that  $ER_{wc}$  is not controlled by a single variable, and instead multiple factors (i.e., nutrient concentrations, suspended particles, and temperature) are simultaneously linked to  $ER_{wc}$ .

Collinearity between LASSO variables could result in one variable being retained in the LASSO model over another. We used LASSO regressions across 100 random seeds, averaging the model coefficients, to help minimize spurious outcomes. This revealed relatively small standard deviations of  $\beta$  coefficients compared to mean  $\beta$  coefficient values, indicating that the four most important variables are consistent across seeds, even when one variable is chosen over another (Table 2). For example, total drainage area was correlated with nutrient concentrations and temperature (Fig. S4), which were retained as more directly explaining variation in ER<sub>wc</sub> in the LASSO regression. Additionally, while total drainage area showed a negative univariate correlation with ER<sub>wc</sub> (Fig. 3b), it showed a slight positive relationship with ER<sub>wc</sub> in the LASSO regression. This suggests that total drainage area likely acts as a proxy for regional watershed processes that influence ER<sub>wc</sub> directly, like nutrients and temperature, rather than a causal

relationship (Caissie, 2006; Manning et al., 2020). Similarly, TDN was strongly correlated with other explanatory variables, such as NO<sub>3</sub>-, Cl<sup>-</sup>, and SO<sub>4</sub><sup>2</sup>-, likely reflecting an increase of agricultural inputs that, in turn, lead to faster ER<sub>wc</sub> through supporting microbial metabolism (Bernot et al., 2010). Including phosphorus data could further improve model performance, as phosphorus is often a limiting factor for microbial growth in freshwater rivers (Carroll 2022). Phosphorus limitation is likely in the Yakima River basin, as more than two-thirds of the phosphorus concentrations were below instrument detection, leading to its exclusion from analysis. These results underscore the importance of interpreting LASSO results within the context of all explanatory variables used in the model, particularly in large, heterogenous catchments.

Table 2.  $\beta$  coefficients from LASSO analyses for explaining ER<sub>wc</sub> across the Yakima River Basin. ER<sub>wc</sub> and all explanatory variables were cube root transformed and standardized as z-scores. LASSO was performed over 100 seeds, and  $\beta$  coefficients for each variable were normalized to the maximum  $\beta$  coefficient in each seed and averaged across all seeds for the reported values. Values of zero indicate that while the variable was included in the model, it was deemed not influential in predicting model outcomes and thus was not assigned a  $\beta$  coefficient.

| Predictor Variable             | Mean β Coefficient | Standard Deviation |
|--------------------------------|--------------------|--------------------|
| TDN                            | -0.96              | 0.11               |
| Temperature                    | -0.62              | 0.15               |
| DOC                            | -0.53              | 0.17               |
| TSS                            | -0.36              | 0.16               |
| $NO_3^-$                       | -0.19              | 0.36               |
| $\mathrm{SO_4}^{2	ext{-}}$     | 0                  | 0                  |
| Normalized DOM Transformations | 0                  | 0                  |
| DIC                            | 0                  | 0                  |
| DOM Transformations            | 0                  | 0                  |
| Total drainage area            | 0.0005             | 0.005              |
| DOM Peaks                      | 0.001              | 0.008              |
| Cl <sup>-</sup>                | 0.13               | 0.27               |
| $\mathbb{R}^2$                 | 0.49               | 0.03               |

437438

Figure 4. Scatter plots of cube root transformed variables that were important in the LASSO regression. Cube root transformed  $ER_{wc}$  is the y-axis for all panels. (a) cube root transformed total dissolved nitrogen (TDN); (b) cube root transformed temperature; (c) cube root transformed dissolved organic carbon (DOC); (d) cube root transformed total suspended solids (TSS). Pearson correlation coefficients (r) are provided on each panel.

Faster ER<sub>wc</sub> with increasing TDN, temperature, DOC, and TSS in the Yakima River basin is expected, as nutrients and temperature can impact variation in stream metabolism (Ardón et al., 2021; Bernot et al., 2010; Honious et al., 2021; Hornbach, 2021; Nakano et al., 2022). In-stream metabolism relies on terrestrially-derived and internally-fixed inputs of DOC, which supports heterotrophic metabolism that degrades and removes organic C inputs through respiration (Hall et al., 2016; Hotchkiss & Hall, 2014; Plont et al., 2022). Faster ER<sub>tot</sub> and ER<sub>wc</sub> have been reported with increases in DOC (Bernot et al., 2010; Ellis et al., 2012). However, elevated DOC does not always correspond to greater ERtot, as discharge and residence time also affect C dynamics (Ulseth et al., 2018). In addition to DOC, suspended sediment can regulate ecosystem metabolism by decoupling ecosystem respiration and GPP through limiting light availability, thereby reducing autochthonous C production, and conversely, by stimulating processing of organic matter through increased surface area (Glover et al., 2019; Honious et al., 2021). The increased surface area of suspended particles in the water column provides microsite habitats for microorganisms (Liu et al., 2013; Ochs et al., 2010), where bacterial production and enzymatic activity is concentrated, contributing substantially to material processing in the water column, particularly in rivers 5th order and higher (Gardner & Doyle, 2018; Reisinger et al., 2015). Nutrient dynamics, particularly N, also influence ecosystem respiration, where elevated N concentrations have been linked to increased ecosystem respiration across stream orders (Benstead et al., 2009; Reisinger et al., 2016, 2021; Rosemond et al., 2015). Nitrogen is a key nutrient for microbial growth and is often a limiting nutrient in freshwater rivers (Carroll, 2022). Consistent with this, we found the fastest ER<sub>wc</sub> at an agriculturally-influenced stream with the greatest TDN and NO<sub>3</sub> concentrations. Elevated nutrient levels at this site likely stimulate microbial respiration, similar to Cross et al. (2022) who found an increase in heterotrophic respiration in response to N enrichment. Moreover, respiratory processes are typically faster at higher temperatures (Pietikäinen et al., 2005), which can shift riverine ecosystems toward heterotrophy (Song et al., 2018). By stimulating microbial respiration, higher temperatures can also amplify the effects of increasing nutrients (Cross et al., 2022). Collectively, we infer that increasing temperature and nutrients, potentially from anthropogenic inputs, are the most likely drivers of ER<sub>wc</sub> in the Yakima River basin. Ultimately, our results emphasize the complex and dynamic roles of the physical, chemical, and biological factors that influence ER<sub>wc</sub> in the Yakima River basin and other similar freshwater ecosystems.

## 4 Conclusions, limitations, and next steps

468

470

Our study shows that ER<sub>wc</sub> rates observed in rivers and streams across the Yakima River basin span published rates from studies conducted in rivers across the CONUS and the Amazon River basin. While this study didn't measure ER<sub>tot</sub>, the observed overlap between ER<sub>wc</sub> and literature ER<sub>tot</sub> show the potential relevance of ER<sub>wc</sub> to overall stream metabolism. We pose that the high variability observed in ER<sub>wc</sub> rates across the basin will likely translate into variable contributions of ER<sub>wc</sub> to ER<sub>tot</sub>, ranging from negligible to potentially dominant. We anticipate that these influences will not vary systematically moving down the stream network as we observed a very weak association between ER<sub>wc</sub> and drainage area across the Yakima River basin. Our results point to more localized control and the LASSO regression specifically indicated that ER<sub>wc</sub> is faster with increasing TDN, stream temperature, DOC, and TSS, consistent with previous work. Overall, our findings show that the complex interactions between physical and chemical factors affect the spatial variability in ER<sub>wc</sub> across the Yakima River basin. We encourage future work to expand on our current study by collecting both ER<sub>wc</sub> and ER<sub>tot</sub> measurements at a basin scale, and to consider areal rates to parse the contributions from both the water column and sediments to total ecosystem metabolism.

#### Code and data availability

- Data and scripts used to generate the main findings within this manuscript will be published at the U.S. Department of Energy's
- Environmental System Science Data Infrastructure for a Virtual Ecosystem (ESS-DIVE) repository (https://ess-
- dive.lbl.gov/about/) upon manuscript acceptance. Currently, scripts associated with this manuscript are located on GitHub
- (https://github.com/river-corridors-sfa/rcsfa-RC2-SPS-ERwc). Other data collected during the field efforts (i.e., sensor data;
- surface water chemistry data; and geospatial information, metadata, and maps for 2021 Spatial Study sampling event) can be
- accessed on ESS-DIVE.

## Supplement link

The persistent DOI and the link to the Supplementary Material will be supplied by EGU Biogeosciences prior to publication.

#### Author contributions

- Conceptualization: JCS, MHK, ROH, SGF, VAGC, MML
- Data Curation: MML, SGF, YF, BF, VAGC, AEG, SG, MHK, XL, AMP, OO, and KS
- Formal Analysis: MML, SGF, VAGC, MHK, XL, AMP, OO, and JT
- Funding Acquisition: XC, TDS, and JCS
- Investigation: MML, SGF, MEB, MAB, VAGC, SG, MHK, XL, SAM, AMP, OO, ACP, HR, LR, KS, JT, KS, JMT, and JCS
- Methodology: MML, SGF, MHK, VAGC, ROH, XL, SAM, AMP, OO, HR, LR, KS, JT, and JCS

- Project Administration: SGF, VAGC, SG, MHK, SAM, AMP, OO, LR, and JCS
- Resources: MML, SGF, VAGC, SG, MHK, SAM, AMP, OO, and LR
- Software: MML, SGF, VAGC, BF, MHK, XL, AMP, KS, and EM
- Supervision: VAGC, XC, MHK, TDS, and JCS
- Validation: MML, SGF, VAGC, SG, MHK, XL, AMP, OO, HR, and JCS
- Visualization: MML, BF, SGF, MHK, XL, SAM, and JT
- Writing Original Draft Preparation: MML, SGF, VAGC, MHK, AMP, JT, and JCS
- Writing Review & Editing: MML, SGF, MEB, VAGC, BF, AEG, ROH, MHK, AMP, KS, JT, and JCS

#### 485 **Competing interest**

The authors declare that they have no conflict of interest.

#### Acknowledgements

- This research was supported by the U.S. Department of Energy (DOE), Office of Science, Office of Biological and Environmental
- Research, Environmental System Science (ESS) Program (https://ess.science.energy.gov/). This contribution originates from the
- River Corridor Scientific Focus Area (SFA) project at Pacific Northwest National Laboratory (PNNL). PNNL is operated by
- Battelle Memorial Institute for the U.S. DOE under Contract No. DE-AC05-76RL01830. FTICR-MS data was generated at the
- DOE BER Environmental Molecular Science Laboratory User Facility (EMSL; https://www.pnnl.gov/environmental-molecular-
- sciences-laboratory) under user proposal 60221. We thank the Confederated Tribes and Bands of the Yakama Nation, US Forest
- Service (USFS), Washington Department of Natural Resources (WDNR), Washington State Parks, Cowiche Canyon Conservancy,
- and Washington Department of Fish and Wildlife (WDFW) for access to field locations where these samples were collected. We
- also thank the Yakama Nation Tribal Council and Yakama Nation Fisheries for working with us to facilitate sample collection and
- optimization of data usage according to their values and worldview. The authors would also like to thank A.J. Reisinger for
- providing water column respiration data included in this study for 13 mid-sized turbid midwestern rivers and western rivers
- (Reisinger et al., 2021) as well as helpful insights and discussions with the lead author on the state-of-the-science on water column
- respiration.

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

| 540 | series. Limnology and Oceanography. https://doi.org/10.1002/lno.12207                                                        |
|-----|------------------------------------------------------------------------------------------------------------------------------|
| 541 | Blodgett, D. L., & Johnson, J. M. (2022). nhdplusTools: Tools for accessing and working with the NHDPlus [Computer           |
| 542 | software]. https://doi.org/10.5066/P97AS8JD                                                                                  |
| 543 | Bramer, L. M., White, A. M., Stratton, K. G., Thompson, A. M., Claborne, D., Hofmockel, K., & McCue, L. A. (2020).           |
| 544 | ftmsRanalysis: An R package for exploratory data analysis and interactive visualization of FT-MS data. PLOS                  |
| 545 | Computational Biology, 16(3), e1007654. https://doi.org/10.1371/journal.pcbi.1007654                                         |
| 546 | Brown, J. (2024). Annual National Land Cover Database (NLCD) Collection 1 Science Product User Guide.                        |
| 547 | https://www.mrlc.gov                                                                                                         |
| 548 | Caissie, D. (2006). The thermal regime of rivers: A review. Freshwater Biology, 51(8), 1389–1406.                            |
| 549 | https://doi.org/10.1111/j.1365-2427.2006.01597.x                                                                             |
| 550 | Carroll, J. (2022). Quality Assurance Project Plan: Lower Yakima River Monitoring for Aquatic Life Parameters to Support     |
| 551 | Water Quality Gaging. (p. 40). Washington State Department of Ecology.                                                       |
| 552 | Cross, W. F., Hood, J. M., Benstead, J. P., Huryn, A. D., Welter, J. R., Gíslason, G. M., & Ólafsson, J. S. (2022). Nutrient |
| 553 | enrichment intensifies the effects of warming on metabolic balance of stream ecosystems. Limnology and                       |
| 554 | Oceanography Letters, 7(4), 332-341. https://doi.org/10.1002/lol2.10244                                                      |
| 555 | Danczak, R. E., Garayburu-Caruso, V. A., Renteria, L., McKever, S. A., Otenburg, O. C., Grieger, S. R., Son, K., Kaufman, M. |
| 556 | H., Fulton, S. G., Roebuck, J. A., Myers-Pigg, A. N., & Stegen, J. C. (2023). Riverine organic matter functional             |
| 557 | diversity increases with catchment size. Frontiers in Water, 5. https://doi.org/10.3389/frwa.2023.1087108                    |
| 558 | del Giorgio, P. A., & Williams, P. J. le B. (2005). Respiration in Aquatic Ecosystems. Oxford University Press.              |
| 559 | Demars, B. O. L. (2019). Hydrological pulses and burning of dissolved organic carbon by stream respiration. Limnology and    |
| 560 | Oceanography, 64(1), 406-421. https://doi.org/10.1002/lno.11048                                                              |
| 561 | Devol, A. H., Forsberg, B. R., Richey, J. E., & Pimentel, T. P. (1995). Seasonal variation in chemical distributions in the  |
| 562 | Amazon (Solimões) River: A multiyear time series. Global Biogeochemical Cycles, 9(3), 307–328.                               |
| 563 | https://doi.org/10.1029/95GB01145                                                                                            |
| 564 | Dillon, G. K. (2018). Wildfire Hazard Potential (WHP) for the conterminous United States (270-m GRID), version 2018          |
| 565 | continuous. Forest Service Research Data Archive.                                                                            |
| 566 | Dillon, G. K., & Gilbertson-Day, J. W. (2020). Wildfire Hazard Potential for the United States (270-m), version 2020.        |
| 567 | Dittmar, T., Koch, B., Hertkorn, N., & Kattner, G. (2008). A simple and efficient method for the solid-phase extraction of   |
| 568 | dissolved organic matter (SPE-DOM) from seawater. Limnology and Oceanography: Methods, 6(6), 230-235.                        |
| 569 | https://doi.org/10.4319/lom.2008.6.230                                                                                       |

- Ellis, E. E., Richey, J. E., Aufdenkampe, A. K., Krusche, A. V., Quay, P. D., Salimon, C., & Da Cunha, H. B. (2012). Factors
- controlling water-column respiration in rivers of the central and southwestern Amazon Basin. *Limnology and*
- Oceanography, 57(2), 527–540. https://doi.org/10.4319/lo.2012.57.2.0527
- Findlay, S. (1995). Importance of surface-subsurface exchange in stream ecosystems: The hyporheic zone. *Limnology and*
- Oceanography, 40(1), 159–164. https://doi.org/10.4319/lo.1995.40.1.0159
- Finlay, J. C. (2011). Stream size and human influences on ecosystem production in river networks. *Ecosphere*, 2(8), art87.
- https://doi.org/10.1890/es11-00071.1
- Friedl, M., & Sulla-Menashe, D. (2019). MCD12Q1 MODIS/Terra+Aqua Land Cover Type Yearly L3 Global 500m SIN Grid
- *V006*.
- Friedman, J. H., Hastie, T., & Tibshirani, R. (2010). Regularization Paths for Generalized Linear Models via Coordinate
- Descent. Journal of Statistical Software, 33(1), 1–22. https://doi.org/10.18637/jss.v033.i01
- Fudyma, J. D., Chu, R. K., Graf Grachet, N., Stegen, J. C., & Tfaily, M. M. (2021). Coupled Biotic-Abiotic Processes Control
- Biogeochemical Cycling of Dissolved Organic Matter in the Columbia River Hyporheic Zone. *Frontiers in Water*, 2.
- https://doi.org/10.3389/frwa.2020.574692
- Fulton, S. G., Barnes, M., Borton, M. A., Chen, X., Farris, Y., Forbes, B., Garayburu-Caruso, V. A., Goldman, A. E., Grieger, S.,
- Kaufman, M. H., Lin, X., McKever, S. A., Myers-Pigg, A., Otenburg, O., Pelly, A., Ren, H., Renteria, L., Scheibe, T.
- D., Son, K., ... Stegen, J. C. (2022). Spatial Study 2021: Sensor-Based Time Series of Surface Water Temperature,
- Specific Conductance, Total Dissolved Solids, Turbidity, pH, and Dissolved Oxygen from across Multiple Watersheds in
- the Yakima River Basin, Washington, USA (v3).
- Fuss, C., & Smock, L. (1996). Spatial and temporal variation of microbial respiration rates in a blackwater stream. Freshwater
- *Biology*, 36(2), 339–349. https://doi.org/10.1046/j.1365-2427.1996.00095.x
- Gagne-Maynard, W. C., Ward, N. D., Keil, R. G., Sawakuchi, H. O., Da Cunha, A. C., Neu, V., Brito, D. C., Da Silva Less, D.
- F., Diniz, J. E. M., De Matos Valerio, A., Kampel, M., Krusche, A. V., & Richey, J. E. (2017). Evaluation of Primary
- Production in the Lower Amazon River Based on a Dissolved Oxygen Stable Isotopic Mass Balance. Frontiers in
- *Marine Science*, 4. https://doi.org/10.3389/fmars.2017.00026
- Garayburu-Caruso, V. A., Danczak, R. E., Stegen, J. C., Renteria, L., McCall, M., Goldman, A. E., Chu, R. K., Toyoda, J.,
- Resch, C. T., Torgeson, J. M., Wells, J., Fansler, S., Kumar, S., & Graham, E. B. (2020a). Using Community Science to
- Reveal the Global Chemogeography of River Metabolomes. *Metabolites*, 10(12), 518.
- https://doi.org/10.3390/metabo10120518
- Garayburu-Caruso, V. A., Stegen, J. C., Song, H.-S., Renteria, L., Wells, J., Garcia, W., Resch, C. T., Goldman, A. E., Chu, R.

| 600 | K., Toyoda, J., & Graham, E. B. (2020b). Carbon Limitation Leads to Thermodynamic Regulation of Aerobic                          |
|-----|----------------------------------------------------------------------------------------------------------------------------------|
| 601 | Metabolism. Environmental Science & Technology Letters, 7(7), 517–524. https://doi.org/10.1021/acs.estlett.0c00258               |
| 602 | Gardner, J. R., & Doyle, M. W. (2018). Sediment-Water Surface Area Along Rivers: Water Column Versus Benthic.                    |
| 603 | Ecosystems, 21(8), 1505–1520. https://doi.org/10.1007/s10021-018-0236-2                                                          |
| 604 | Genzoli, L., & Hall, R. O. (2016). Shifts in Klamath River metabolism following a reservoir cyanobacterial bloom. Freshwater     |
| 605 | Science, 35(3), 795–809. https://doi.org/10.1086/687752                                                                          |
| 606 | Glover, H. E., Ogston, A. S., Miller, I. M., Eidam, E. F., Rubin, S. P., & Berry, H. D. (2019). Impacts of Suspended Sediment or |
| 607 | Nearshore Benthic Light Availability Following Dam Removal in a Small Mountainous River: In Situ Observations and                |
| 608 | Statistical Modeling. Estuaries and Coasts, 42(7), 1804–1820. https://doi.org/10.1007/s12237-019-00602-5                         |
| 609 | Gomez-Velez, J. D., Harvey, J. W., Cardenas, M. B., & Kiel, B. (2015). Denitrification in the Mississippi River network          |
| 610 | controlled by flow through river bedforms. Nature Geoscience, 8(12), 941–945. https://doi.org/10.1038/ngeo2567                   |
| 611 | Grieger, S., Barnes, M., Borton, M. A., Chen, X., Chu, R., Farris, Y., Forbes, B., Fulton, S. G., Garayburu-Caruso, V. A.,       |
| 612 | Goldman, A. E., Gonzalez, B. I., Kaufman, M. H., McKever, S. A., Myers-Pigg, A., Otenburg, O., Pelly, A., Renteria,              |
| 613 | L., Scheibe, T. D., Son, K., Stegen, J. C. (2022). Spatial Study 2021: Sample-Based Surface Water Chemistry and                  |
| 614 | Organic Matter Characterization across Watersheds in the Yakima River Basin, Washington, USA (v3) (U. S. DOE >                   |
| 615 | Office of Science > Biological and Environmental Research (BER), Trans.) [Surface water chemistry sampling].                     |
| 616 | Hall, R. O. (2016). Chapter 4 - Metabolism of Streams and Rivers: Estimation, Controls, and Application. In J. B. Jones & E. H.  |
| 617 | Stanley (Eds.), Stream Ecosystems in a Changing Environment (pp. 151-180). Academic Press.                                       |
| 618 | https://doi.org/10.1016/B978-0-12-405890-3.00004-X                                                                               |
| 619 | Hall, R. O., & Hotchkiss, E. R. (2017). Chapter 34—Stream Metabolism. In G. A. Lamberti & F. R. Hauer (Eds.), Methods in         |
| 620 | Stream Ecology (Third Edition) (pp. 219–233). Academic Press.                                                                    |
| 621 | Hall, R. O., Tank, J. L., Baker, M. A., Rosi-Marshall, E. J., & Hotchkiss, E. R. (2016). Metabolism, Gas Exchange, and Carbon    |
| 622 | Spiraling in Rivers. <i>Ecosystems</i> , 19(1), 73–86. https://doi.org/10.1007/s10021-015-9918-1                                 |
| 623 | Hensley, R. T., Kirk, L., Spangler, M., Gooseff, M. N., & Cohen, M. J. (2019). Flow Extremes as Spatiotemporal Control Points    |
| 624 | on River Solute Fluxes and Metabolism. Journal of Geophysical Research: Biogeosciences, 124(3), 537–555.                         |
| 625 | https://doi.org/10.1029/2018jg004738                                                                                             |

Honious, S. A. S., Hale, R. L., Guilinger, J. J., Crosby, B. T., & Baxter, C. V. (2021). Turbidity Structures the Controls of 626 627 Ecosystem Metabolism and Associated Metabolic Process Domains Along a 75-km Segment of a Semiarid Stream. 628 *Ecosystems*. https://doi.org/10.1007/s10021-021-00661-5

Hornbach, D. J. (2021). Multi-Year Monitoring of Ecosystem Metabolism in Two Branches of a Cold-Water Stream.

| 630 | Environments, 8(3), 19.                                                                                                                                                                                                                                                                                                                                                                                                                                                                                                                                                                                                                                                                                                                                                                                                                                                                                                                                                                                                                                                                                                                                                                                                                                                                                                                                                                                                                                                                                                                                                                                                                                                                                                                                                                                                                                                                                                                                                                                                                                                                                                       |
|-----|-------------------------------------------------------------------------------------------------------------------------------------------------------------------------------------------------------------------------------------------------------------------------------------------------------------------------------------------------------------------------------------------------------------------------------------------------------------------------------------------------------------------------------------------------------------------------------------------------------------------------------------------------------------------------------------------------------------------------------------------------------------------------------------------------------------------------------------------------------------------------------------------------------------------------------------------------------------------------------------------------------------------------------------------------------------------------------------------------------------------------------------------------------------------------------------------------------------------------------------------------------------------------------------------------------------------------------------------------------------------------------------------------------------------------------------------------------------------------------------------------------------------------------------------------------------------------------------------------------------------------------------------------------------------------------------------------------------------------------------------------------------------------------------------------------------------------------------------------------------------------------------------------------------------------------------------------------------------------------------------------------------------------------------------------------------------------------------------------------------------------------|
| 631 | Hotchkiss, E. R., & Hall, R. O. (2014). High rates of daytime respiration in three streams: Use of δ 18 OO2 and O2 to model die                                                                                                                                                                                                                                                                                                                                                                                                                                                                                                                                                                                                                                                                                                                                                                                                                                                                                                                                                                                                                                                                                                                                                                                                                                                                                                                                                                                                                                                                                                                                                                                                                                                                                                                                                                                                                                                                                                                                                                                               |
| 632 | ecosystem metabolism. Limnology and Oceanography, 59(3), 798-810. https://doi.org/10.4319/lo.2014.59.3.0798                                                                                                                                                                                                                                                                                                                                                                                                                                                                                                                                                                                                                                                                                                                                                                                                                                                                                                                                                                                                                                                                                                                                                                                                                                                                                                                                                                                                                                                                                                                                                                                                                                                                                                                                                                                                                                                                                                                                                                                                                   |
| 633 | Jankowski, K. J., & Schindler, D. E. (2019). Watershed geomorphology modifies the sensitivity of aquatic ecosystem                                                                                                                                                                                                                                                                                                                                                                                                                                                                                                                                                                                                                                                                                                                                                                                                                                                                                                                                                                                                                                                                                                                                                                                                                                                                                                                                                                                                                                                                                                                                                                                                                                                                                                                                                                                                                                                                                                                                                                                                            |
| 634 | metabolism to temperature. Sci Rep, 9(1), 17619. https://doi.org/10.1038/s41598-019-53703-3                                                                                                                                                                                                                                                                                                                                                                                                                                                                                                                                                                                                                                                                                                                                                                                                                                                                                                                                                                                                                                                                                                                                                                                                                                                                                                                                                                                                                                                                                                                                                                                                                                                                                                                                                                                                                                                                                                                                                                                                                                   |
| 635 | Jones Jr, J. b. (1995). Factors controlling hyporheic respiration in a desert stream. Freshwater Biology, 34(1), 91–99.                                                                                                                                                                                                                                                                                                                                                                                                                                                                                                                                                                                                                                                                                                                                                                                                                                                                                                                                                                                                                                                                                                                                                                                                                                                                                                                                                                                                                                                                                                                                                                                                                                                                                                                                                                                                                                                                                                                                                                                                       |
| 636 | https://doi.org/10.1111/j.1365-2427.1995.tb00426.x                                                                                                                                                                                                                                                                                                                                                                                                                                                                                                                                                                                                                                                                                                                                                                                                                                                                                                                                                                                                                                                                                                                                                                                                                                                                                                                                                                                                                                                                                                                                                                                                                                                                                                                                                                                                                                                                                                                                                                                                                                                                            |
| 637 | Kaplan, L. A., & Newbold, J. D. (2000). 10—Surface and Subsurface Dissolved Organic Carbon. In J. B. Jones & P. J.                                                                                                                                                                                                                                                                                                                                                                                                                                                                                                                                                                                                                                                                                                                                                                                                                                                                                                                                                                                                                                                                                                                                                                                                                                                                                                                                                                                                                                                                                                                                                                                                                                                                                                                                                                                                                                                                                                                                                                                                            |
| 638 | Mulholland (Eds.), Streams and Ground Waters (pp. 237-258). Academic Press. https://doi.org/10.1016/B978-                                                                                                                                                                                                                                                                                                                                                                                                                                                                                                                                                                                                                                                                                                                                                                                                                                                                                                                                                                                                                                                                                                                                                                                                                                                                                                                                                                                                                                                                                                                                                                                                                                                                                                                                                                                                                                                                                                                                                                                                                     |
| 639 | 012389845-6/50011-9                                                                                                                                                                                                                                                                                                                                                                                                                                                                                                                                                                                                                                                                                                                                                                                                                                                                                                                                                                                                                                                                                                                                                                                                                                                                                                                                                                                                                                                                                                                                                                                                                                                                                                                                                                                                                                                                                                                                                                                                                                                                                                           |
| 640 | Liu, T., Xia, X., Liu, S., Mou, X., & Qiu, Y. (2013). Acceleration of Denitrification in Turbid Rivers Due to Denitrification                                                                                                                                                                                                                                                                                                                                                                                                                                                                                                                                                                                                                                                                                                                                                                                                                                                                                                                                                                                                                                                                                                                                                                                                                                                                                                                                                                                                                                                                                                                                                                                                                                                                                                                                                                                                                                                                                                                                                                                                 |
| 641 | Occurring on Suspended Sediment in Oxic Waters. Environmental Science & Environmental & Environmental & Environmental & Environmental & Environmental |
| 642 | https://doi.org/10.1021/es304504m                                                                                                                                                                                                                                                                                                                                                                                                                                                                                                                                                                                                                                                                                                                                                                                                                                                                                                                                                                                                                                                                                                                                                                                                                                                                                                                                                                                                                                                                                                                                                                                                                                                                                                                                                                                                                                                                                                                                                                                                                                                                                             |
| 643 | Manning, D. W. P., Rosemond, A. D., Benstead, J. P., Bumpers, P. M., & Kominoski, J. S. (2020). Transport of N and P in U.S.                                                                                                                                                                                                                                                                                                                                                                                                                                                                                                                                                                                                                                                                                                                                                                                                                                                                                                                                                                                                                                                                                                                                                                                                                                                                                                                                                                                                                                                                                                                                                                                                                                                                                                                                                                                                                                                                                                                                                                                                  |
| 644 | streams and rivers differs with land use and between dissolved and particulate forms. Ecological Applications, 30(6),                                                                                                                                                                                                                                                                                                                                                                                                                                                                                                                                                                                                                                                                                                                                                                                                                                                                                                                                                                                                                                                                                                                                                                                                                                                                                                                                                                                                                                                                                                                                                                                                                                                                                                                                                                                                                                                                                                                                                                                                         |
| 645 | e02130. https://doi.org/10.1002/eap.2130                                                                                                                                                                                                                                                                                                                                                                                                                                                                                                                                                                                                                                                                                                                                                                                                                                                                                                                                                                                                                                                                                                                                                                                                                                                                                                                                                                                                                                                                                                                                                                                                                                                                                                                                                                                                                                                                                                                                                                                                                                                                                      |
| 646 | McKay, L., Bondelid, T., Dewald, T., Johnston, J., Moore, R., & Rea, A. (2012). NHDPlus Version 2: User Guide.                                                                                                                                                                                                                                                                                                                                                                                                                                                                                                                                                                                                                                                                                                                                                                                                                                                                                                                                                                                                                                                                                                                                                                                                                                                                                                                                                                                                                                                                                                                                                                                                                                                                                                                                                                                                                                                                                                                                                                                                                |
| 647 | Mulholland, P. J., Helton, A. M., Poole, G. C., Hall, R. O., Hamilton, S. K., Peterson, B. J., Tank, J. L., Ashkenas, L. R., Cooper                                                                                                                                                                                                                                                                                                                                                                                                                                                                                                                                                                                                                                                                                                                                                                                                                                                                                                                                                                                                                                                                                                                                                                                                                                                                                                                                                                                                                                                                                                                                                                                                                                                                                                                                                                                                                                                                                                                                                                                           |
| 648 | L. W., Dahm, C. N., Dodds, W. K., Findlay, S. E. G., Gregory, S. V., Grimm, N. B., Johnson, S. L., McDowell, W. H.,                                                                                                                                                                                                                                                                                                                                                                                                                                                                                                                                                                                                                                                                                                                                                                                                                                                                                                                                                                                                                                                                                                                                                                                                                                                                                                                                                                                                                                                                                                                                                                                                                                                                                                                                                                                                                                                                                                                                                                                                           |
| 649 | Meyer, J. L., Valett, H. M., Webster, J. R., Thomas, S. M. (2008). Stream denitrification across biomes and its                                                                                                                                                                                                                                                                                                                                                                                                                                                                                                                                                                                                                                                                                                                                                                                                                                                                                                                                                                                                                                                                                                                                                                                                                                                                                                                                                                                                                                                                                                                                                                                                                                                                                                                                                                                                                                                                                                                                                                                                               |
| 650 | response to anthropogenic nitrate loading. Nature, 452(7184), 202–205. https://doi.org/10.1038/nature06686                                                                                                                                                                                                                                                                                                                                                                                                                                                                                                                                                                                                                                                                                                                                                                                                                                                                                                                                                                                                                                                                                                                                                                                                                                                                                                                                                                                                                                                                                                                                                                                                                                                                                                                                                                                                                                                                                                                                                                                                                    |
| 651 | Myneni, R., Knyazikhin, Y., & Park, T. (2015). MCD15A3H MODIS/Terra+Aqua Leaf Area Index/FPAR 4-day L4 Global 500m                                                                                                                                                                                                                                                                                                                                                                                                                                                                                                                                                                                                                                                                                                                                                                                                                                                                                                                                                                                                                                                                                                                                                                                                                                                                                                                                                                                                                                                                                                                                                                                                                                                                                                                                                                                                                                                                                                                                                                                                            |
| 652 | SIN Grid V006.                                                                                                                                                                                                                                                                                                                                                                                                                                                                                                                                                                                                                                                                                                                                                                                                                                                                                                                                                                                                                                                                                                                                                                                                                                                                                                                                                                                                                                                                                                                                                                                                                                                                                                                                                                                                                                                                                                                                                                                                                                                                                                                |
| 653 | Naegeli, M. W., & Uehlinger, U. (1997). Contribution of the Hyporheic Zone to Ecosystem Metabolism in a Prealpine Gravel-                                                                                                                                                                                                                                                                                                                                                                                                                                                                                                                                                                                                                                                                                                                                                                                                                                                                                                                                                                                                                                                                                                                                                                                                                                                                                                                                                                                                                                                                                                                                                                                                                                                                                                                                                                                                                                                                                                                                                                                                     |
| 654 | Bed-River. Journal of the North American Benthological Society, 16(4), 794–804. https://doi.org/10.2307/1468172                                                                                                                                                                                                                                                                                                                                                                                                                                                                                                                                                                                                                                                                                                                                                                                                                                                                                                                                                                                                                                                                                                                                                                                                                                                                                                                                                                                                                                                                                                                                                                                                                                                                                                                                                                                                                                                                                                                                                                                                               |
| 655 | Nakano, D., Iwata, T., Suzuki, J., Okada, T., Yamamoto, R., & Imamura, M. (2022). The effects of temperature and light on                                                                                                                                                                                                                                                                                                                                                                                                                                                                                                                                                                                                                                                                                                                                                                                                                                                                                                                                                                                                                                                                                                                                                                                                                                                                                                                                                                                                                                                                                                                                                                                                                                                                                                                                                                                                                                                                                                                                                                                                     |

ecosystem metabolism in a Japanese stream. *Freshwater Science*, *41*(1), 113–124. https://doi.org/10.1086/718648
Ochs, C. A., Capello, H. E., & Pongruktham, O. (2010). Bacterial production in the Lower Mississippi River: Importance of
suspended sediment and phytoplankton biomass. *Hydrobiologia*, *637*(1), 19–31. https://doi.org/10.1007/s10750-009-

- Peterson, B. J., Wollheim, W. M., Mulholland, P. J., Webster, J. R., Meyer, J. L., Tank, J. L., Martí, E., Bowden, W. B., Valett,
- H. M., Hershey, A. E., McDowell, W. H., Dodds, W. K., Hamilton, S. K., Gregory, S., & Morrall, D. D. (2001). Control
- of Nitrogen Export from Watersheds by Headwater Streams. Science, 292(5514), 86–90.
- https://doi.org/doi:10.1126/science.1056874
- Pietikäinen, J., Pettersson, M., & Bååth, E. (2005). Comparison of temperature effects on soil respiration and bacterial and fungal
- growth rates. FEMS Microbiology Ecology, 52(1), 49–58. https://doi.org/10.1016/j.femsec.2004.10.002
- Plont, S., Riney, J., & Hotchkiss, E. R. (2022). Integrating Perspectives on Dissolved Organic Carbon Removal and Whole-
- Stream Metabolism. *Journal of Geophysical Research-Biogeosciences*, 127(3). https://doi.org/10.1029/2021JG006610
- PRISM Climate Group, Oregon State University. (2023, June 29). https://prism.oregonstate.edu
- Quay, P. D., Wilbur, D., Richey, J. E., Devol, A. H., Benner, R., & Forsberg, B. R. (1995). The 18O:16O of dissolved oxygen in
- rivers and lakes in the Amazon Basin: Determining the ratio of respiration to photosynthesis rates in freshwaters.
- Limnology and Oceanography, 40(4), 718–729. https://doi.org/10.4319/lo.1995.40.4.0718
- Reisinger, A. J., Tank, J. L., Hall, R. O., Rosi, E. J., Baker, M. A., & Genzoli, L. (2021). Water column contributions to the
- metabolism and nutrient dynamics of mid-sized rivers. *Biogeochemistry*, 153(1), 67–84. https://doi.org/10.1007/s10533-
- 021-00768-w
- Reisinger, A. J., Tank, J. L., Hoellein, T. J., & Hall, R. O. (2016). Sediment, water column, and open-channel denitrification in
- rivers measured using membrane-inlet mass spectrometry. Journal of Geophysical Research: Biogeosciences, 121(5),
- 1258–1274. https://doi.org/10.1002/2015jg003261
- Reisinger, A. J., Tank, J. L., Rosi-Marshall, E. J., Hall, R. O., & Baker, M. A. (2015). The varying role of water column nutrient
- uptake along river continua in contrasting landscapes. *Biogeochemistry*, 125(1), 115–131.
- https://doi.org/10.1007/s10533-015-0118-z
- Rosemond, A. D., Benstead, J. P., Bumpers, P. M., Gulis, V., Kominoski, J. S., Manning, D. W. P., Suberkropp, K., & Wallace,
- J. B. (2015). Experimental nutrient additions accelerate terrestrial carbon loss from stream ecosystems. *Science*,
- 347(6226), 1142–1145. https://doi.org/10.1126/science.aaa1958
- Running, S., Mu, Q., & Zhao, M. (2017). MOD16A2 MODIS/Terra Net Evapotranspiration 8-Day L4 Global 500m SIN Grid
- *V006*.
- Running, S., & Zhao, M. (2019). MOD17A3HGF MODIS/Terra Net Primary Production Gap-Filled Yearly L4 Global 500 m
- *SIN Grid V006*.
- Ryan, K. A., Garayburu-Caruso, V. A., Crump, B. C., Bambakidis, T., Raymond, P. A., Liu, S., & Stegen, J. C. (2024). Riverine
- dissolved organic matter transformations increase with watershed area, water residence time, and Damköhler numbers

| 690 |  |  | 0.1007/s10533-024-01169-5 |
|-----|--|--|---------------------------|
|     |  |  |                           |

- Song, C., Dodds, W. K., Rüegg, J., Argerich, A., Baker, C. L., Bowden, W. B., Douglas, M. M., Farrell, K. J., Flinn, M. B.,
- Garcia, E. A., Helton, A. M., Harms, T. K., Jia, S., Jones, J. B., Koenig, L. E., Kominoski, J. S., McDowell, W. H.,
- McMaster, D., Parker, S. P., ... Ballantyne, F. (2018). Continental-scale decrease in net primary productivity in streams
- due to climate warming. *Nature Geoscience*, 11(6), 415–420. https://doi.org/10.1038/s41561-018-0125-5
- Stegen, J. C., Johnson, T., Fredrickson, J. K., Wilkins, M. J., Konopka, A. E., Nelson, W. C., Arntzen, E. V., Chrisler, W. B.,
- Chu, R. K., Fansler, S. J., Graham, E. B., Kennedy, D. W., Resch, C. T., Tfaily, M., & Zachara, J. (2018). Influences of
- organic carbon speciation on hyporheic corridor biogeochemistry and microbial ecology. *Nature Communications*, 9(1),
- 585. https://doi.org/10.1038/s41467-018-02922-9
- Tolić, N., Liu, Y., Liyu, A., Shen, Y., Tfaily, M. M., Kujawinski, E. B., Longnecker, K., Kuo, L.-J., Robinson, E. W., Paša-
- Tolić, L., & Hess, N. J. (2017). Formularity: Software for Automated Formula Assignment of Natural and Other
- Organic Matter from Ultrahigh-Resolution Mass Spectra. *Analytical Chemistry*, 89(23), 12659–12665.
- https://doi.org/10.1021/acs.analchem.7b03318
- U. S. Geological Survey. (2019). C6 Aqua 250-m eMODIS Remote Sensing Phenology Metrics across the conterminous U.S.
- Ulseth, A. J., Bertuzzo, E., Singer, G. A., Schelker, J., & Battin, T. J. (2018). Climate-Induced Changes in Spring Snowmelt
- Impact Ecosystem Metabolism and Carbon Fluxes in an Alpine Stream Network. *Ecosystems*, 21(2), 373–390.
- https://doi.org/10.1007/s10021-017-0155-7
- U.S Geological Survey. (2023, June 29). National Elevation Dataset (NED) 1/3 Arc-Second Digital Elevation Model.
- http://nationalmap.gov/elevation.html
- Vano, J. A., Scott, M. J., Voisin, N., Stöckle, C. O., Hamlet, A. F., Mickelson, K. E. B., Elsner, M. M., & Lettenmaier, D. P.
- (2010). Climate change impacts on water management and irrigated agriculture in the Yakima River Basin,
- Washington, USA. Climatic Change, 102(1), 287–317. https://doi.org/10.1007/s10584-010-9856-z
- Ward, N. D., Sawakuchi, H. O., Neu, V., Less, D. F. S., Valerio, A. M., Cunha, A. C., Kampel, M., Bianchi, T. S., Krusche, A.
- 713 V., Richey, J. E., & Keil, R. G. (2018). Velocity-amplified microbial respiration rates in the lower Amazon River.
- Limnology and Oceanography Letters, 3(3), 265–274. https://doi.org/10.1002/lol2.10062
- Webster, J. R. (2007). Spiraling down the river continuum: Stream ecology and the U-shaped curve. *Journal of the North*
- American Benthological Society, 26(3), 375–389. https://doi.org/10.1899/06-095.1
- Willi, K., & Ross, M. R. V. (2023). Geospatial Data Puller for Waters in the Contiguous United States (Version v1) [Zenodo].
- https://doi.org/10.5281/zenodo.8140272