# Peer review of "Water Column Respiration in the Yakima River Basin is Explained by"

_EGUsphere, 2025_

## Referee Comment (RC1)

**Review of "Water Column Respiration in the Yakima River Basin is Explained by Temperature, Nutrients and Suspended Solids" by Laan et al.**

**General comments:**

Laan et al. submitted a well-written manuscript with a clear structure. The key strengths of this work are that Laan et al. could reliably measure ecosystem respiration in the water column ($ER_{WC}$) at a broad spatial scale in environmentally diverse rivers, which are located at different positions in the Yakima River network and that they measured a comprehensive set of explanatory variables for $ER_{WC}$ ranging from basic physicochemical measures to high-resolution DOM data.

Laan et al. hypothesized that $ER_{WC}$ values increase from upstream to downstream (approximated by drainage area) and conclude from their results that rather local factors are driving $ER_{WC}$ in river networks. Although I, in general, agree with the conclusion, I do not think that the data analysis is robust enough to really rule out the effect of drainage area (and regional processes in general) on $ER_{WC}$ because of two potential problems:

1. Laan et al. use LASSO regressions for variable selection and regularization. LASSO analysis does not provide p-values per se, however, one could argue that only important variables are selected in the model. Total drainage area was selected in the final model with a low positive mean beta coefficient suggesting a slight increase of $ER_{WC}$ values (i.e. rates slow down) from upstream to downstream. However, based on Fig. 3b, I see a decreasing trend of $ER_{WC}$ (i.e. rates fasten) from upstream to downstream. I suspect that this discrepancy is due to strong effects of other variables on $ER_{WC}$, which also strongly correlate with drainage area (Fig. S4). I suggest to try partial regression plots to illustrate the net effects of predictors on $ER_{WC}$. For Fig. 3b this could mean that you show how $ER_{WC}$ changed as a function of the drainage area after statistically holding the effect of other important predictors (e.g. TDN) constant. This strong collinearity among predictors (Fig. S4), however, may have even more influence on the analysis: If there are highly correlated predictors, LASSO's variable selection might only select one of them. I suggest to provide an explanation why you do not think that this is an issue for your analysis or a comparison of your current method with methods using elastic net penalties (combination of penalties of the LASSO and ridge method) which may deal better with multicollinearity.

2. The model does not account for spatial autocorrelation, which could further lead to an overseen stronger shift of $ER_{WC}$ from upstream to downstream. Either you directly use spatially explicit methods for your analysis, such as spatial stream network models (*e.g. Peterson, E. E. & Ver Hoef, J. M. A mixed-model moving-average approach to geostatistical modeling in stream networks. Ecology **91**, 644–651 (2010)*) or you discuss this issue more thoroughly. For example, when looking at TDN along the drainage area (Fig. S4), I assume that TDN concentrations increase from upstream to downstream along two major branches of the network, which finally merge into the most downstream sites, suggesting a strong spatial autocorrelation of TDN

concentrations. Hence, regional processes (transport of water chemical from upstream to downstream) might have an indirect effect on $ER_{WC}$ by shaping the spatial TDN pattern across the entire river network.

Further, I suggest to change negative $ER_{WC}$ values to positive values (consumption of O2), as I think it makes the interpretation of positive and negative beta coefficients in the LASSO analysis easier and it is more common in the literature. This would also solve ambiguity when writing "$ER_{WC}$ increases", because in these cases I was often not sure if the authors mean whether $ER_{WC}$ rates increase (i.e. rates fasten) or $ER_{WC}$ values increase (i.e. rates slow down).

Overall, considering that my major concerns and specific comments will be addressed in a major revision, this study warrants publication in Biogeosciences.

**Specific comments:**

26: Reading about positive ER values in the Abstract is very confusing. After reading the method section I understand that you kept positive values below 0.5 because they are difficult to distinguish from zero. Still, I wonder if you could avoid stating positive ER values in the Abstract to avoid confusion by the readers early on. But also see my specific comment to line 292.

26: Although you clearly state that you "did not test this directly", I think it is still too much to say "that the contribution of $ER_{WC}$ rates to reach-scale $ER_{tot}$ rates across the Yakima River basin are likely highly variable", because a comparison of a set of $ER_{WC}$ values with another set of $ER_{tot}$ values, which both derive from a wide range of streams differing in size, stream order, environment, etc. does not give you that information. You might come closer to such a result if you apply the same classification to the streams from the literature as you have done to your study streams and then compare $ER_{WC}$ and $ER_{tot}$ within classes.

28: I suggest to keep the statement that you "did not observe a clear **increase** in $ER_{WC}$" more neutral. Because at that point of the manuscript, it is not clear why you expect an increase in absolute $ER_{WC}$ values from upstream to downstream. But see also my next specific comment.

71: I miss an explanation why you expect **absolute** $ER_{WC}$ values to increase. At the beginning of this paragraph you state that water column processes are expected to become increasingly important from upstream to downstream due to a shift from benthic-dominated processes to water column dominated processes. This statement is about the **relative** contribution of water column processes. However, I do not think that this easily translates to your hypothesis that **absolute** $ER_{WC}$ values will increase from upstream to downstream, because $ER_{tot}$ is expected to decrease (rates slow down) from upstream to downstream (e.g. *Segatto, P.L., Battin, T.J. & Bertuzzo, E. (2021). The Metabolic Regimes at the Scale of an Entire Stream Network Unveiled Through Sensor Data and Machine Learning. Ecosystems, 24, 1792–1809.*).

148: Please write out the abbreviation "DO" first time you mention it.

200: Can you please clarify how calculating "the distance between each of the replicate samples" helps to identify the outlier.

255: perform instead of performs

292: I wonder, whether keeping positive $ER_{WC}$ values below 0.5 in your data biases your results (Fig. 2) towards low respiration rates. Maybe this is the reason why you find on average lower respiration rates compared to studies from the literature. Since you are comparing your data to the literature, it might be reasonable to investigate how studies from literature were dealing with positive $ER_{WC}$ values and subsequently use the same approach. I definitely do not suggest to erase cases with positive $ER_{WC}$ values below 0.5 from the results but I find it more intuitive to change positive $ER_{WC}$ values below 0.5 to zeros, as you rightfully stated that positive values are biologically unrealistic and are not distinguishable from zero. However, if studies from the literature research also keep positive values, I would suggest to keep the current approach to make results comparable.

342: I understood that you use LASSO analyses to investigate which variables are selected for your final model. Hence, a rejection of your hypothesis at this point of the manuscript based on the correlation in Fig. 3b seems premature. See also my general comment to this topic

347: This part of the sentence is not clear: "… as opposed to a higher order river", please clarify.

364: Change to "Regression analyses show**ed** …"

**Technical corrections**

88: Remove space previous to dot.

216: The reference is missing at the end of the sentence.

---

## Author Comment (AC1)

**Reviewer 1: Thank you for your helpful comments and the time you put into reviewing the paper. Below, our responses are provided in bolded text following each reviewer comment. The responses are primarily in the form of our plan to address each comment in a revised manuscript, if afforded that opportunity by the editor. We look forward to your further evaluation.**

General comments:
Laan et al. submitted a well-written manuscript with a clear structure. The key strengths of this work are that Laan et al. could reliably measure ecosystem respiration in the water column (ERWC) at a broad spatial scale in environmentally diverse rivers, which are located at different positions in the Yakima River network and that they measured a comprehensive set of explanatory variables for ERWC ranging from basic physicochemical measures to high-resolution DOM data.

**Thank you for the encouraging remarks.**

Laan et al. hypothesized that ERWC values increase from upstream to downstream (approximated by drainage area) and conclude from their results that rather local factors are driving ERWC in river networks. Although I, in general, agree with the conclusion, I do not think that the data analysis is robust enough to really rule out the effect of drainage area (and regional processes in general) on ERWC because of two potential problems:

1. Laan et al. use LASSO regressions for variable selection and regularization. LASSO analysis does not provide p-values per se, however, one could argue that only important variables are selected in the model. Total drainage area was selected in the final model with a low positive mean beta coefficient suggesting a slight increase of ERWC values (i.e. rates slow down) from upstream to downstream. However, based on Fig. 3b, I see a decreasing trend of ERWC (i.e.rates fasten) from upstream to downstream. I suspect that this discrepancy is due to strong effects of other variables on ERWC, which also strongly correlate with drainage area (Fig. S4). I suggest to try partial regression plots to illustrate the net effects of predictors on ERWC. For Fig. 3b this could mean that you show how ERWC changed as a function of the drainage area after statistically holding the effect of other important predictors (e.g. TDN) constant. This strong collinearity among predictors (Fig. S4), however, may have even more influence on the analysis: If there are highly correlated predictors, LASSO's variable selection might only select one of them. I suggest to provide an explanation why you do not think that this is an issue for your analysis or a comparison of your current method with methods using elastic net penalties (combination of penalties of the LASSO and ridge method) which may deal better with multicollinearity.

**In a revised manuscript we plan to address the high-level point of collinearity influences by adding more acknowledgement of the collinearity among explanatory variables. We will emphasize relationships between LASSO-identified variables and ion concentrations**

that seem to indicate influences of agricultural inputs. We plan to retain our approach of emphasizing N, rather than TDN or $NO_3$, as to avoid assumptions about which of those variables is most important. We will keep our approach of refraining from placing an order of importance to the variables and maintain a focus on each as explaining $ER_{wc}$ across the Yakima River basin.

To address the more specific point about total drainage area, we made the following five observations and plan to take actions summarized in each bullet.

1) To help minimize spurious outcomes that could arise due to collinearity across parameters, we used LASSO regressions across 100 random seeds, averaging the model coefficients. This reduced the likelihood of arbitrarily selecting one variable over another, which could happen with high collinearity. This revealed relatively small standard deviations of $\beta$ coefficients compared to mean $\beta$ coefficient values, indicating that the most important variables are consistent across seeds. In a revised manuscript, we plan to add a small amount of additional text to help make these points clearer.

2) In preliminary analyses, not currently shown in the paper, we tested the sensitivity of our LASSO results to collinearity by removing highly correlated explanatory variables. In this case, we had to select between co-correlated variables and include one of them in the LASSO. We selected the variable to keep based on it having the higher (univariate) Pearson correlation coefficient with $ER_{wc}$. This did not change identities of the most important explanatory variables, as identified via LASSO. This consistency suggests that collinearity did not meaningfully alter the key findings from the LASSO modeling. Including these analyses in the revised manuscript does not feel necessary, but we can include these analyses in the supplementary material if the editor deems it necessary; we will await guidance on this point.

3) The variables identified as the most important via LASSO also had the largest Pearson correlation coefficients with $ER_{wc}$. This consistency between multivariate and univariate methods provides additional confidence in the robustness of the LASSO outcomes. This is already discussed in the paper and we plan to retain that text.

4) A partial regression based on total drainage area, controlling for all the variables that had non-zero $\beta$ coefficients in the LASSO model, is shown below. The relationship between total drainage area and $ER_{wc}$ remains weak in this partial regression plot. Additionally, the direction of the relationship is opposite from the univariate correlation; $ER_{wc}$ was faster moving downstream in the univariate analysis but slower moving downstream in the LASSO model. We do not plan to add the partial regression plot to the manuscript, but will add more discussion of

**how the relationship between total drainage area and other explanatory variables may relate to ER$_{wc}$ as described below.**

[Figure]

**Added-Variable Plot: scale_cube_TotDr**

5)  The highest Pearson correlation of total drainage area was r = 0.63 with temperature, followed by an r = 0.58 correlation with TDN. Temperature and TDN are indicated by the LASSO model as being important for explaining variation in ER$_{wc}$. Clear mechanistic connections between ER$_{wc}$ and both temperature and TDN suggest that any relationship between drainage area and ER$_{wc}$ is not causal. Instead, we infer that the relationship between total drainage area and ER$_{wc}$ is simply because of correlations among the explanatory variables, and not a mechanistic connection. This means that drainage area itself is not the key factor, but instead may act as a proxy for influences of temperature and TDN. Further, TDN is strongly correlated with other explanatory variables, including NO$_3$, Cl, and SO$_4$.. The covariation among these variables likely reflects increasing influences of agricultural inputs that, in turn, lead to faster ER$_{wc}$ by supporting microbial metabolism. We collectively infer that increasing temperature and nutrients, potentially from agricultural inputs, are the most likely drivers of ER$_{wc}$ rates in the YRB.  In a revised manuscript, we plan to add more discussion of these points.

2a. The model does not account for spatial autocorrelation, which could further lead to an overseen stronger shift of ERWC from upstream to downstream. Either you directly use spatially explicit methods for your analysis, such as spatial stream network models (e.g. Peterson, E. E. & Ver Hoef, J. M. A mixed-model moving-average approach to geostatistical modeling in stream networks. Ecology 91, 644–651 (2010)) or you discuss this issue more thoroughly. For example, when looking at TDN along the drainage area (Fig. S4), I assume that TDN concentrations increase from upstream to downstream along two major branches of the network, which finally merge into the most downstream sites, suggesting a strong spatial autocorrelation of TDN concentrations. Hence, regional processes (transport of water chemical from upstream to downstream) might have an indirect effect on ERWC by shaping the spatial TDN pattern across the entire river network.

**We plan to add discussion on the possibility of regional watershed processes influencing key explanatory variables such as total dissolved nitrogen, as suggested by the reviewer. We find this very helpful and we certainly agree that the chemistry observed at a given site is the outcome of processes occurring throughout the upstream drainage area (watershed).**

2b. Further, I suggest to change negative ERWC values to positive values (consumption of O2), as I think it makes the interpretation of positive and negative beta coefficients in the LASSO analysis easier and it is more common in the literature. This would also solve ambiguity when writing "ERWC increases", because in these cases I was often not sure if the authors mean whether ERWC rates increase (i.e. rates fasten) or ERWC values increase (i.e. rates slow down). Overall, considering that my major concerns and specific comments will be addressed in a major revision, this study warrants publication in Biogeosciences.

**We appreciate the suggestion to express all values as positive to more closely match literature representations and improve ambiguity. To clarify interpretation in the text, we will replace the occurrences of "rates increase" with revised language in lines 28, 71, and 74 to explicitly state that we are referring to rate magnitudes (i.e., faster or slower). However, we plan to retain the sign of $ER_{wc}$ values to reduce bias that would be introduced by setting small positive values, likely associated with instrument noise, to zero. As described in lines 245 - 252 and 291 - 294, these small positive values represent measurement uncertainty overwhelming the rate of oxygen consumption, which means the rate of oxygen consumption must have been very slow. Retaining the sign allows us to preserve this information across the full data set, where more negative values indicate faster oxygen consumption.**

Specific comments:
26: Reading about positive ER values in the Abstract is very confusing. After reading the method section I understand that you kept positive values below 0.5 because they are

difficult to distinguish from zero. Still, I wonder if you could avoid stating positive ER values in the Abstract to avoid confusion by the readers early on. But also see my specific comment to line 292.

**We will improve clarity by presenting results in the Abstract as something similar to "...rates ranged from effectively zero to -7.38..."**

26: Although you clearly state that you "did not test this directly", I think it is still too much to say "that the contribution of ERWC rates to reach-scale ERtot rates across the Yakima River basin are likely highly variable", because a comparison of a set of ERWC values with another set of ERtot values, which both derive from a wide range of streams differing in size, stream order, environment, etc. does not give you that information. You might come closer to such a result if you apply the same classification to the streams from the literature as you have done to your study streams and then compare ERWC and ERtot within classes.

**To clarify, we plan to revise the results/discussion text to emphasize that this comparison is exploratory and intended to assess the potential overlap between our measured $ER_{wc}$ values and $ER_{tot}$ reported in the literature. While both datasets span a range of stream conditions, we acknowledge that differences in environmental context limits direct comparability. Nevertheless, the overlap between $ER_{wc}$ and $ER_{tot}$ values suggests that $ER_{wc}$ could contribute a variable portion of $ER_{tot}$, with some instances indicating the potential for substantial contributions. Substantial contributions of $ER_{wc}$ to $ER_{tot}$ are not guaranteed, however, which we will emphasize in the revised manuscript. The results fail to reject the hypothesis that $ER_{wc}$ can be a significant portion of $ER_{tot}$. If we had observed consistently very slow $ER_{wc}$ rates across the Yakima River basin, we would see effectively no overlap with $ER_{tot}$ values. This would have clearly rejected the hypothesis that $ER_{wc}$ can contribute substantially to $ER_{tot}$.**

**We also agree that a more robust analysis would involve applying consistent stream classifications across datasets and comparing $ER_{wc}$ and $ER_{tot}$ within those classes. We will revise the manuscript to more clearly state the limitations of this initial comparison, while still noting that it provides useful context for understanding the range of $ER_{wc}$ values observed in the YRB.**

28: I suggest to keep the statement that you "did not observe a clear increase in ERWC" more neutral. Because at that point of the manuscript, it is not clear why you expect an increase in absolute ERWC values from upstream to downstream. But see also my next specific comment.

**We plan to change this sentence to: We were able to explain 40% of ERwc variability via a combination of temperature, dissolved organic carbon, total dissolved nitrogen, and total suspended solids.**

71: I miss an explanation why you expect absolute ERWC values to increase. At the beginning of this paragraph you state that water column processes are expected to become

increasingly important from upstream to downstream due to a shift from benthic-dominated processes to water column dominated processes. This statement is about the relative contribution of water column processes. However, I do not think that this easily translates to your hypothesis that absolute ERWC values will increase from upstream to downstream, because ERtot is expected to decrease (rates slow down) from upstream to downstream (e.g. Segatto, P.L., Battin, T.J. & Bertuzzo, E. (2021). The Metabolic Regimes at the Scale of an Entire Stream Network Unveiled Through Sensor Data and Machine Learning. Ecosystems, 24, 1792–1809.).

**We plan to add additional material to the introduction to provide more rationale for the hypothesis that $ER_{wc}$ increases moving downstream. For example, we plan to highlight the increase in gross primary production (GPP) (Marzolf and Ardon, 2021; Segatto et al., 2021) moving down the stream network. Increases in GPP are known to influence ecosystem respiration, such that we would expect an increase in $ER_{wc}$ moving downstream. Additionally, increases in nitrogen processing in the water column (Wang et al., 2022), may also indicate that $ER_{wc}$ would increase with stream order.**

**Marzolf, N.S., and Ardón, M. (2021). Ecosystem metabolism in tropical streams and rivers: a review and synthesis. *Limnology and Oceanography, 66* (5), 1627 - 1638.**

**Segatto, P.L., Battin, T.J., and Bertuzzo, E. (2021). The Metabolic Regimes at the Scale of an Entire Stream Network Unveiled Through Sensor Data and Machine Learning. *Ecosystems, 24,* 1792 - 1809.**

**Wang, J., Xia, X., Liu, S., Zhang, S., Zhang, L., Jiang, C., Zhang, Z., Xin, Y., Chen, X., Huang, J., Bao, J., McDowell, W.H., Michalski, G., Yang, Z., and Xia, J. (2022). The Dominant Role of the Water Column in Nitrogen Removal and N2O Emissions in Large Rivers. *Geophysical Research Letters, 49* (12).**

148: Please write out the abbreviation "DO" first time you mention it.

**We plan to add "dissolved oxygen" at line 148.**

200: Can you please clarify how calculating "the distance between each of the replicate samples" helps to identify the outlier.

**We plan to add the following information to the methods section describing this calculation for identifying outliers. Pairwise differences between NPOC, TDN, and DIC measurements from all replicates were calculated. The sample that had the largest difference from the other samples was removed if the coefficient of variation was greater than 30%. This coefficient of variation threshold for sample removal is based on inspecting histograms of these data types, and determining the point at which sites likely contain anomalous outlier values.**

255: perform instead of performs

**We plan to change this.**

292: I wonder, whether keeping positive ERWC values below 0.5 in your data biases your results (Fig. 2) towards low respiration rates. Maybe this is the reason why you find on average lower respiration rates compared to studies from the literature. Since you are comparing your data to the literature, it might be reasonable to investigate how studies from literature were dealing with positive ERWC values and subsequently use the same approach. I definitely do not suggest to erase cases with positive ERWC values below 0.5 from the results but I find it more intuitive to change positive ERWC values below 0.5 to zeros, as you rightfully stated that positive values are biologically unrealistic and are not distinguishable from zero. However, if studies from the literature research also keep positive values, I would suggest to keep the current approach to make results comparable.

**To the best of our knowledge, other published studies of $ER_{wc}$ do not discuss positive rates, so there is no precedent to base our decision off of. We are left with needing to set the precedent. We make the assumption that variation among near-zero positive $ER_{wc}$ rates does carry some real information such that keeping those rates carries that information into the statistical analyses. If we change all near-zero positive $ER_{wc}$ rates to zero, we lose that information. In turn, we prefer to retain the near-zero $ER_{wc}$ rates and in the revised manuscript we plan to further emphasize the assumption we're making.**

**Regarding the influence of this decision on our comparison to literature values of $ER_{wc}$, in our view the main take away from our study is that within a single basin the range of $ER_{wc}$ is broader than all literature values combined. This outcome won't be altered by changing near-zero positive $ER_{wc}$ rates to zero. This further motivates us to retain the near-zero positive $ER_{wc}$ rates in a revised manuscript.**

342: I understood that you use LASSO analyses to investigate which variables are selected for your final model. Hence, a rejection of your hypothesis at this point of the manuscript based on the correlation in Fig. 3b seems premature. See also my general comment to this topic

**We plan to soften the language at this point in the manuscript to acknowledge this point. We will edit the text to something like "...is inconsistent with our hypothesis and below we use a multivariate analysis for further evaluation."**

347: This part of the sentence is not clear: "... as opposed to a higher order river", please clarify.

**We plan to edit this sentence to clarify that we expected ER$_{wc}$ to be fastest in the highest order rivers, not mid-order rivers, by changing the text to something like "... as opposed to our hypothesis of ER$_{wc}$ being fastest in the highest stream orders."**

364: Change to "Regression analyses showed ..."

**We will change this in the revised manuscript.**

Technical corrections
88: Remove space previous to dot.
**We will change this in the revised manuscript.**

216: The reference is missing at the end of the sentence.
**We will change this in the revised manuscript.**

---

## Author Comment (AC2)

**Reviewer 2: Thank you for your helpful comments and the time you put into reviewing the paper. Below, our responses are provided in bolded text following each reviewer comment. The responses are primarily in the form of our plan to address each comment in a revised manuscript, if afforded that opportunity by the editor. We look forward to your further evaluation.**

Laan et al. evaluated water column respiration (ERwc) and water quality parameters at 47 sites in the Yakima River Basin. The goal of the study was to identify factors driving changes in ERwc throughout the river network using LASSO regressions. In addition, the authors collected total and ERwc values from other studies in the continental US and the Amazon River basin. In general, the authors found no clear increase in ERwc over the course of the river network, and ERwc rates were influenced by local factors such as temperature, dissolved organic carbon, total dissolved nitrogen, and suspended sediment, rather than position in the stream network. In addition, the range of ERwc in the Yakima River Basin encompassed the entire range of ERwc that the authors found in the other studies, and ERwc contributed differentially to ERtot from the other studies.

**Thank you for the encouraging remarks.**

This study is well-focused and addresses a question regarding the processes occurring in the water column of river networks. The research is thorough and directed, leaving little room for criticism from my perspective. I'll leave two comments here that I would have liked the authors to address in a little more detail to see if/how relevant this might be to their study.

One point I am thinking about is the discussion of the importance of water column and sediment processes to overall metabolism. In line 22 of the abstract and several times throughout the manuscript, the authors state that "the relative influence of sediment-associated processes versus water column processes can fluctuate along the river continuum." In my opinion, an important factor in this statement is the greater influence of water column processes due to higher water levels when going downstream, which increases the areal influence of the water column. However, the authors compare volumetric rates, which do not consider the influence of water column height. Why did the authors decide to compare volumetric values? I'm not criticizing the approach, but I think the theory they are testing is largely based on this relationship. This could be a point that could (or should?) be included in the discussion.

**One reason we looked at volumetric rates is that we did not have access to high quality depth data for all the field sites where we estimated ER$_{wc}$. To get good depth data would be a major effort in the Yakima River Basin. Some locations are small streams (relatively easy to get depth via manual measurements) while other locations are on the 7th order main stem (much harder to get depth). In addition, some literature estimates of ER$_{wc}$ are in volumetric units and no depth data are provided; the only way to do a direct comparison across all literature rates is via volumetric units. Nonetheless, we acknowledge the importance of considering water column depth for understanding variation in the contribution of water column processes to whole system respiration (i.e.,**

**ER$_{tot}$). To address this in the revised manuscript we plan to clarify why we used volumetric rates and also add some discussion on the value of also considering water column depth, per the reviewer's comments.**

The authors state in line 395 that "Nitrogen is a key nutrient for microbial growth and is often a limiting nutrient in freshwater rivers (Carroll, 2022)." Another common limiting factor is phosphorus. The authors use a variety of water and catchment parameters to perform the regression. However, phosphorus was not examined. Is there a reason for this? Is this not a potential important factor for ecosystem metabolism in the Yakima River Basin? Including this factor could improve the significance of the regression and significantly influence the conclusion that 40% can be predicted.

**We agree that phosphorus is often limiting and probably is important in the Yakima River Basin. Our analysis of phosphorus showed values below detection for more than two-thirds of samples, which is further evidence that it probably is limiting. Because of the analytical limitations, we feel there isn't enough good phosphorus data to include in the analyses. This is unfortunate of course. To address this in the revised manuscript we plan to acknowledge that phosphorus is likely a key nutrient and that while we attempted to measure it, we didn't get data of sufficient quality to include them in the analysis. We will also point out that if we had those data, they would likely explain further variation in ERwc, and we will provide encouragement for future studies to include phosphorus in ERwc studies.**

Minor comments:

Line 30: "...which explained 40% of ERwc variability across the basin." You could add here the method you used to come to this number as you use LASSO regression, which has certain assumptions.

**LASSO does not natively provide an R-squared estimate, but it does allow for predicting values of the response variable based on values of the explanatory variables. This, in turn, allows estimation of the residual sum of squares (RSS). The total sum of squares (TSS) does not depend on the regression model or predictions and can be directly estimated. We estimated R-squared for each of the 100 LASSO models as 1 - RSS/TSS, as traditionally done with standard multiple regression. We emphasize that we did not use the resulting R-squared estimates as part of the model estimation process, but rather as a way to estimate how much variation in ERwc was explained by each of the 100 LASSO models. We also computed mean and standard deviation of the R-squared values across the 100 LASSO models. In the revised manuscript we will provide a summary of these points to clarify our approach.**

Line 216: Reference missing

**We will include the reference in the revised manuscript.**

Line 390: Could not find Ochs et al. 2010 in the reference list

**We will make sure this reference is included in the revised manuscript's reference list.**

---

## Author Response (AR1)

Dear Editor and Reviewers.

Thank you for the helpful review of our manuscript. We have completed the revision, per your suggestions. Our revisions are summarized below, with reviewer text in standard font and our responses in bold font. We appreciate your time and look forward to your further evaluation.

Sincerely,

James Stegen (on behalf of all co-authors)

###

**Reviewer 1:**

**General comments:**

Laan et al. submitted a well-written manuscript with a clear structure. The key strengths of this work are that Laan et al. could reliably measure ecosystem respiration in the water column (ERWC) at a broad spatial scale in environmentally diverse rivers, which are located at different positions in the Yakima River network and that they measured a comprehensive set of explanatory variables for ERWC ranging from basic physicochemical measures to high-resolution DOM data.

**Thank you for the encouraging remarks.**

Laan et al. hypothesized that ERWC values increase from upstream to downstream (approximated by drainage area) and conclude from their results that rather local factors are driving ERWC in river networks. Although I, in general, agree with the conclusion, I do not think that the data analysis is robust enough to really rule out the effect of drainage area (and regional processes in general) on ERWC because of two potential problems:

1. Laan et al. use LASSO regressions for variable selection and regularization. LASSO analysis does not provide p-values per se, however, one could argue that only important variables are selected in the model. Total drainage area was selected in the final model with a low positive mean beta coefficient suggesting a slight increase of ERWC values (i.e. rates slow down) from upstream to downstream. However, based on Fig. 3b, I see a decreasing trend of ERWC (i.e.rates fasten) from upstream to downstream. I suspect that this discrepancy is due to strong effects of other variables on ERWC, which also strongly correlate with drainage area (Fig. S4). I suggest to try partial regression plots to illustrate the net effects of predictors on ERWC. For Fig. 3b this could mean that you show how ERWC changed as a function of the drainage area after statistically holding the effect of other important predictors (e.g. TDN) constant. This strong collinearity among predictors (Fig. S4), however, may have even more influence on the analysis: If there are highly correlated predictors, LASSO's variable selection might only select one of them. I suggest to provide an explanation why you do not think that this is an issue for your analysis or a

comparison of your current method with methods using elastic net penalties (combination of penalties of the LASSO and ridge method) which may deal better with multicollinearity.

In the revised manuscript, we addressed the high-level point of collinearity influences on parameters chosen in the LASSO regression in the second paragraph of section 3.3. We emphasized relationships between LASSO-identified variables and correlated ion concentrations that seem to indicate agricultural influences. We retained our approach of emphasizing N, rather than TDN or  $NO_3$ , in paragraph 3 of section 3.3 to avoid assumptions about which of those variables is most important. We also refrained from placing an order of importance to the variables to maintain a focus on each as explaining  $ER_{wc}$  across the Yakima River basin.

To address the more specific point about total drainage area, we made the following five observations and plan to take actions summarized in each bullet.

- 1) To help minimize spurious outcomes that could arise due to collinearity across parameters, we used LASSO regressions across 100 random seeds, averaging the model coefficients. This reduced the likelihood of arbitrarily selecting one variable over another, which could happen with high collinearity. This revealed relatively small standard deviations of  $\beta$  coefficients compared to mean  $\beta$  coefficient values, indicating that the four most important variables are consistent across seeds, even when one variable is chosen over another. In the revised manuscript, we added more text explaining this comparison between seeds in section 2.7.
- 2) In preliminary analyses, not shown in the paper, we tested the sensitivity of our LASSO results to collinearity by removing highly correlated explanatory variables. In this case, we had to select between co-correlated variables and include one of them in the LASSO. We selected the variable to keep based on it having the higher (univariate) Pearson correlation coefficient with ERwc. This did not change identities of the most important explanatory variables, as identified via LASSO. This consistency suggests that collinearity did not meaningfully alter the key findings from the LASSO modeling. These analyses are not included in the revised manuscript.
- 3) The variables identified as the most important via LASSO also had the largest Pearson correlation coefficients with  $ER_{wc}$ . This consistency between multivariate and univariate methods provides additional confidence in the robustness of the LASSO outcomes. This is already discussed in the first paragraph of section 3.3 of the paper and we retained that text.
- 4) A partial regression based on total drainage area, controlling for all the variables that had non-zero  $\beta$  coefficients in the LASSO model, is shown below. The relationship between total drainage area and ERwc remains weak in this partial

regression plot. Additionally, the direction of the relationship is opposite from the univariate correlation;  $ER_{wc}$  was faster moving downstream in the univariate analysis but slower moving downstream in the LASSO model. We do not plan to add the partial regression plot to the manuscript, but added more discussion of how the relationship between total drainage area and other explanatory variables may relate to  $ER_{wc}$  as described below.

5) The highest Pearson correlation of total drainage area was r = 0.63 with temperature, followed by an r = 0.58 correlation with TDN. Temperature and TDN are indicated by the LASSO model as being important for explaining variation in ERwc. Clear mechanistic connections between ERwc and both temperature and TDN suggest that any relationship between drainage area and ERwc is not causal. Instead, we infer that the relationship between total drainage area and ERwc is simply because of correlations among the explanatory variables, and not a

mechanistic connection. This means that drainage area itself is not the key factor, but instead may act as a proxy for influences of temperature and TDN. Further, TDN is strongly correlated with other explanatory variables, including  $NO_3$ , CI, and  $SO_4$ .. The covariation among these variables likely reflects increasing influences of agricultural inputs that, in turn, lead to faster  $ER_{wc}$  by supporting microbial metabolism. We collectively infer that increasing temperature and nutrients, potentially from agricultural inputs, are the most likely drivers of  $ER_{wc}$  rates in the Yakima River basin. In the revised manuscript, we added more discussion of these points, including addressing the sign change between analyses, in the second paragraph of section 3.3.

2a. The model does not account for spatial autocorrelation, which could further lead to an overseen stronger shift of ERWC from upstream to downstream. Either you directly use spatially explicit methods for your analysis, such as spatial stream network models (e.g. Peterson, E. E. & Ver Hoef, J. M. A mixed-model moving-average approach to geostatistical modeling in stream networks. Ecology 91, 644–651 (2010)) or you discuss this issue more thoroughly. For example, when looking at TDN along the drainage area (Fig. S4), I assume that TDN concentrations increase from upstream to downstream along two major branches of the network, which finally merge into the most downstream sites, suggesting a strong spatial autocorrelation of TDN concentrations. Hence, regional processes (transport of water chemical from upstream to downstream) might have an indirect effect on ERWC by shaping the spatial TDN pattern across the entire river network.

We added more discussion on the possibility of regional watershed processes influencing key explanatory variables such as total dissolved nitrogen to the second paragraph of section 3.3. We find this very helpful and we certainly agree that the chemistry observed at a given site is the outcome of processes occurring throughout the upstream drainage area (watershed).

2b. Further, I suggest to change negative ERWC values to positive values (consumption of O2), as I think it makes the interpretation of positive and negative beta coefficients in the LASSO analysis easier and it is more common in the literature. This would also solve ambiguity when writing "ERWC increases", because in these cases I was often not sure if the authors mean whether ERWC rates increase (i.e. rates fasten) or ERWC values increase (i.e. rates slow down). Overall, considering that my major concerns and specific comments will be addressed in a major revision, this study warrants publication in Biogeosciences.

To clarify interpretation in the text, we removed the occurrence of "rates increase" in the abstract, and replaced the occurrences of "rates increase" with revised language in the last paragraph of section 1 to explicitly state that we are referring to rate magnitudes (i.e., faster or slower). All language was changed to "faster or slower" to avoid ambiguity. Slightly positive ERwc values likely stem from very slow oxygen consumption and are

associated with instrument noise. However, at the advice of the reviewer, we changed these slightly positive values to 0 and redid analyses. This did not change the conceptual interpretation of the story. Section 2.6 in the methods and analyses throughout the results and discussion have been updated to reflect this. We retained the negative sign of the values, where more negative values indicate faster oxygen consumption. The main reason to retain this approach is to enable easy comparison to our original results, which are now in the supplementary information. Those original results include the slightly positive ERwc rates such that we can't simply change the sign of the rates.

**Specific comments:**

26: Reading about positive ER values in the Abstract is very confusing. After reading the method section I understand that you kept positive values below 0.5 because they are difficult to distinguish from zero. Still, I wonder if you could avoid stating positive ER values in the Abstract to avoid confusion by the readers early on. But also see my specific comment to line 292.

We improved clarity of results in the Abstract by changing this sentence to "... 0 to -7.38..."

26: Although you clearly state that you "did not test this directly", I think it is still too much to say "that the contribution of ERWC rates to reach-scale ERtot rates across the Yakima River basin are likely highly variable", because a comparison of a set of ERWC values with another set of ERtot values, which both derive from a wide range of streams differing in size, stream order, environment, etc. does not give you that information. You might come closer to such a result if you apply the same classification to the streams from the literature as you have done to your study streams and then compare ERWC and ERtot within classes.

To clarify that this comparison was exploratory and intended to assess the potential overlap between our measured  $ER_{wc}$  values and  $ER_{tot}$  reported in the literature, we added a sentence describing these limitations in the third paragraph of section 3.1. In this section, we also addressed that the overlap in  $ER_{wc}$  measured in the Yakima River basin and literature  $ER_{tot}$  suggests that  $ER_{wc}$  could contribute a variable amount to  $ER_{tot}$ , which may be substantial in some cases, though this is not guaranteed.

28: I suggest to keep the statement that you "did not observe a clear increase in ERWC" more neutral. Because at that point of the manuscript, it is not clear why you expect an increase in absolute ERWC values from upstream to downstream. But see also my next specific comment.

We changed this sentence to: We observed that  $ER_{wc}$  is locally controlled by temperature, dissolved organic carbon, total dissolved nitrogen, and total suspended solids, which explained 40% of  $ER_{wc}$  variability across the basin using Least Absolute Shrinkage and Selection Operator (LASSO) regression.

71: I miss an explanation why you expect absolute ERWC values to increase. At the beginning

of this paragraph you state that water column processes are expected to become increasingly important from upstream to downstream due to a shift from benthic-dominated processes to water column dominated processes. This statement is about the relative contribution of water column processes. However, I do not think that this easily translates to your hypothesis that absolute ERWC values will increase from upstream to downstream, because ERtot is expected to decrease (rates slow down) from upstream to downstream (e.g. Segatto, P.L., Battin, T.J. & Bertuzzo, E. (2021). The Metabolic Regimes at the Scale of an Entire Stream Network Unveiled Through Sensor Data and Machine Learning. Ecosystems, 24, 1792–1809.).

We added the following additional material to the third paragraph of the Introduction to provide more rationale for the hypothesis that  $ER_{wc}$  increases moving downstream:

"Increases in downstream GPP (Finlay, 2011; Segatto et al., 2021), may also influence ecosystem respiration, such that we would expect faster  $ER_{wc}$  with greater GPP due to increases in autochthonous C (Hall et al., 2016; Mejia et al., 2019). Additionally, greater N processing in the water column with increasing stream order (Wang et al., 2022), may suggest that water column biogeochemical processing increases along the stream network."

Finlay, J. C. (2011). Stream size and human influences on ecosystem production in river networks. Ecosphere, 2(8), art87. https://doi.org/10.1890/es11-00071.1

Segatto, P.L., Battin, T.J., and Bertuzzo, E. (2021). The Metabolic Regimes at the Scale of an Entire Stream Network Unveiled Through Sensor Data and Machine Learning. *Ecosystems*, 24, 1792 - 1809.

Wang, J., Xia, X., Liu, S., Zhang, S., Zhang, L., Jiang, C., Zhang, Z., Xin, Y., Chen, X., Huang, J., Bao, J., McDowell, W.H., Michalski, G., Yang, Z., and Xia, J. (2022). The Dominant Role of the Water Column in Nitrogen Removal and N2O Emissions in Large Rivers. *Geophysical Research Letters*, 49 (12).

148: Please write out the abbreviation "DO" first time you mention it.

We added "dissolved oxygen" at the first use of DO in the first paragraph of section 2.3.

200: Can you please clarify how calculating "the distance between each of the replicate samples" helps to identify the outlier.

We added the following information in the second paragraph of section 2.4: "Pairwise differences between NPOC, TDN, and DIC measurements from all replicates were calculated. The sample that had the largest difference from the other samples was removed if the coefficient of variation was greater than 30%. This coefficient of variation

threshold for sample removal is based on inspecting histograms of these data types, and determining the point at which sites likely contain anomalous outlier values."

255: perform instead of performs

**This has been updated.**

292: I wonder, whether keeping positive ERWC values below 0.5 in your data biases your results (Fig. 2) towards low respiration rates. Maybe this is the reason why you find on average lower respiration rates compared to studies from the literature. Since you are comparing your data to the literature, it might be reasonable to investigate how studies from literature were dealing with positive ERWC values and subsequently use the same approach. I definitely do not suggest to erase cases with positive ERWC values below 0.5 from the results but I find it more intuitive to change positive ERWC values below 0.5 to zeros, as you rightfully stated that positive values are biologically unrealistic and are not distinguishable from zero. However, if studies from the literature research also keep positive values, I would suggest to keep the current approach to make results comparable.

To the best of our knowledge, other published studies of ERwc do not discuss positive rates, so there is no precedent to base our decision off of. However, we have changed the slightly positive values measured across the Yakima River basin to a value of zero at the reviewer's suggestion. This did not change the conceptual interpretation of the analysis. This has been updated in section 2.6 in the Methods, Figure 2, Figure 3, Figure 4, and Table 2.

342: I understood that you use LASSO analyses to investigate which variables are selected for your final model. Hence, a rejection of your hypothesis at this point of the manuscript based on the correlation in Fig. 3b seems premature. See also my general comment to this topic

We updated this language in the first paragraph of section 3.2 to include that the relationship "...between  $ER_{wc}$  and total drainage area across the Yakima River basin was weak enough that we consider it inconsistent with our hypothesis..." and added text clarifying that multivariate analysis is used in subsequent sections for further evaluation.

347: This part of the sentence is not clear: "... as opposed to a higher order river", please clarify.

We edited this sentence to clarify that the results of the fastest observed  $ER_{wc}$  occurring in a 5th order stream was "... opposed to our hypothesis of  $ER_{wc}$  being fastest in the highest stream orders."

364: Change to "Regression analyses showed ..."

**This has been updated in the revised manuscript.**

Technical corrections

88: Remove space previous to dot.

This has been updated in the revised manuscript.

216: The reference is missing at the end of the sentence.

This has been updated in the revised manuscript.

**Reviewer 2:**

Laan et al. evaluated water column respiration (ERwc) and water quality parameters at 47 sites in the Yakima River Basin. The goal of the study was to identify factors driving changes in ERwc throughout the river network using LASSO regressions. In addition, the authors collected total and ERwc values from other studies in the continental US and the Amazon River basin. In general, the authors found no clear increase in ERwc over the course of the river network, and ERwc rates were influenced by local factors such as temperature, dissolved organic carbon, total dissolved nitrogen, and suspended sediment, rather than position in the stream network. In addition, the range of ERwc in the Yakima River Basin encompassed the entire range of ERwc that the authors found in the other studies, and ERwc contributed differentially to ERtot from the other studies.

**Thank you for the encouraging remarks.**

This study is well-focused and addresses a question regarding the processes occurring in the water column of river networks. The research is thorough and directed, leaving little room for criticism from my perspective. I'll leave two comments here that I would have liked the authors to address in a little more detail to see if/how relevant this might be to their study.

One point I am thinking about is the discussion of the importance of water column and sediment processes to overall metabolism. In line 22 of the abstract and several times throughout the manuscript, the authors state that "the relative influence of sediment-associated processes versus water column processes can fluctuate along the river continuum." In my opinion, an important factor in this statement is the greater influence of water column processes due to higher water levels when going downstream, which increases the areal influence of the water column. However, the authors compare volumetric rates, which do not consider the influence of water column height. Why did the authors decide to compare volumetric values? I'm not criticizing the approach, but I think the theory they are testing is largely based on this relationship. This could be a point that could (or should?) be included in the discussion.

One reason we used volumetric rates for analysis is the difficulties in obtaining high quality depth data at all sites across the Yakima River basin. This study includes 7th order main stem sites, where it would be challenging to measure depth. This has been clarified in section 2.6 of the Methods. In addition, some literature estimates of  $ER_{wc}$  are in volumetric units and no depth data are provided; the only way to do a direct comparison across all literature rates is via volumetric units. Nonetheless, we acknowledge the importance of considering water column depth for understanding variation in the contribution of water column processes to whole system respiration (i.e.,  $ER_{tot}$ ) in the first paragraph of section 3.2. We also encourage future studies to measure water column depth to obtain areal  $ER_{wc}$  rates in the Conclusion.

The authors state in line 395 that "Nitrogen is a key nutrient for microbial growth and is often a limiting nutrient in freshwater rivers (Carroll, 2022)." Another common limiting factor is phosphorus. The authors use a variety of water and catchment parameters to perform the regression. However, phosphorus was not examined. Is there a reason for this? Is this not a potential important factor for ecosystem metabolism in the Yakima River Basin? Including this factor could improve the significance of the regression and significantly influence the conclusion that 40% can be predicted.

We agree that phosphorus is often limiting and probably is important in the Yakima River Basin. Our analysis of phosphorus showed values below detection for more than two-thirds of samples, which is further evidence that it probably is limiting. Because of the analytical limitations, we feel there isn't enough good phosphorus data to include in the analyses. This is unfortunate of course. To address this in the revised manuscript, we acknowledged that phosphorus is likely a key nutrient in the second paragraph of section 3.3 and that while we attempted to measure it, we didn't get data of sufficient quality to include them in the analysis (second paragraph of section 2.4).

**Minor comments:**

Line 30: "...which explained 40% of ERwc variability across the basin." You could add here the method you used to come to this number as you use LASSO regression, which has certain assumptions.

We described LASSO regression as the method used for regression analyses in the Abstract per the reviewers suggestion. We also added the following description of how R2 was calculated in section 2.7:

"LASSO does not inherently estimate  $R^2$ , so we calculated it using the total sum of squares and residual sum of squares for each fitted model, as traditionally done with standard multiple regression. The estimation of residual sum of squares used predicted values of  $ER_{wc}$  based on the explanatory variables used in the model. The  $R^2$  estimates were used to estimate how much variation in  $ER_{wc}$  was explained by each of the LASSO models. Standard deviation of  $\beta$  coefficients were compared to mean values of  $\beta$

coefficients to confirm that the most important variables were relatively consistent across seeds.".

Line 216: Reference missing

The missing reference has been added.

We will include the reference in the revised manuscript.

Line 390: Could not find Ochs et al. 2010 in the reference list

The missing reference has been added.

---

## Referee Report (RR1)

Second review of "Water Column Respiration in the Yakima River Basin is Explained by Temperature, Nutrients and Suspended Solids" by Laan et al.

The revision addresses the main points that I raised in my review of the first submission of this paper and I appreciate the work the authors have put in to clarify and improve the manuscript. I have only one more concern:

I do not agree with line 374 and 376 where the authors state that the correlation between ERWC and total drainage area is too weak to confirm their hypothesis, i.e. ERWC fastens downstream. Because I actually see a descent relationship of ERWC with total drainage area (Figure 3b). This is underlined by a Pearson correlation coefficient of -0.39 and a p-value below 0.05 (Figure S5). In fact, the relationship seems to be curvilinear saturating and therefore the Pearson correlation coefficient, which specifically measures linear correlation and does not capture non-linear relationships such as those represented in Figure 3b, might even underestimate the strength of the relationship. Such non-linear relationships are also visible in Figure 4d and to some extent in Figure 4a. A log transformation of the predictors instead of a cube root transformation might "linearize" those relationships. I apologize for not pointing this out already in the first review. Having said that, I agree with the authors' general conclusion "that localized factors, not upstream conditions or drainage area, provide primary controls over ERWC", as well as I appreciate the added more detailed discussion starting in line 414, still, I think in general ERWC seems to accelerate towards downstream locations, despite strong local controls.

---

## Author Response (AR2)

Dear Dr. Singer,

Thank you for the further evaluation of our manuscript. We're glad to see the reviewers are nearly satisfied. We've addressed the remaining comment in our response below, with our responses in bold text. Thank you for the opportunity to provide this additional revision.

We appreciate your time and look forward to your further thoughts, James Stegen (on behalf of all co-authors)

####

**Reviewer 2:**

The revision addresses the main points that I raised in my review of the first submission of this paper and I appreciate the work the authors have put in to clarify and improve the manuscript. I have only one more concern:

I do not agree with line 374 and 376 where the authors state that the correlation between ERWC and total drainage area is too weak to confirm their hypothesis, i.e. ERWC fastens downstream. Because I actually see a descent relationship of ERWC with total drainage area (Figure 3b). This is underlined by a Pearson correlation coefficient of -0.39 and a p-value below 0.05 (Figure S5). In fact, the relationship seems to be curvilinear saturating and therefore the Pearson correlation coefficient, which specifically measures linear correlation and does not capture non-linear relationships such as those represented in Figure 3b, might even underestimate the strength of the relationship. Such non-linear relationships are also visible in Figure 4d and to some extent in Figure 4a. A log transformation of the predictors instead of a cube root transformation might "linearize" those relationships. I apologize for not pointing this out already in the first review. Having said that, I agree with the authors' general conclusion "that localized factors, not upstream conditions or drainage area, provide primary controls over ERWC", as well as I appreciate the added more detailed discussion starting in line 414, still, I think in general ERWC seems to accelerate towards downstream locations, despite strong local controls.

We're glad to see the reviewer agrees with our ultimate inference and, as such, we haven't altered that part of the manuscript. We edited the section the reviewer refers to, which is section 3.2. The text was edited to acknowledge that there appears to be a relationship, as the reviewer points out. Our view is that the multivariate analyses offer a more robust evaluation of the relative importance of the relationship between ERwc and drainage area. The text already conveyed that perspective, and we edited slightly to help further emphasize this idea. Please see the first paragraph of section 3.2 for the revised text. We also made minor edits to the last paragraph in section 1 and to the single paragraph of section 4. Those additional edits were made to reflect the changes to section 3.2 (i.e., to maintain consistency across the manuscript).